# Geometry Processing with Neural Fields

**Guandao Yang** *
Cornell University

**Serge Belongie**
University of Copenhagen

**Bharath Hariharan**
Cornell University

**Vladlen Koltun**
Intel Labs

## Abstract

Most existing geometry processing algorithms use meshes as the default shape representation. Manipulating meshes, however, requires one to maintain high quality in the surface discretization. For example, changing the topology of a mesh usually requires additional procedures such as remeshing. This paper instead proposes the use of *neural fields* for geometry processing. Neural fields can compactly store complicated shapes without spatial discretization. Moreover, neural fields are infinitely differentiable, which allows them to be optimized for objectives that involve higher-order derivatives. This raises the question: *can geometry processing be done entirely using neural fields?* We introduce loss functions and architectures to show that some of the most challenging geometry processing tasks, such as deformation and filtering, can be done with neural fields. Experimental results show that our methods are on par with the well-established mesh-based methods without committing to a particular surface discretization. Code is available at https://github.com/stevenygd/NFGP.

## 1 Introduction

In many graphics applications, users may want to edit digital shapes using just a few clicks, such as making a character bow by dragging the head downwards. Such manipulation from sparse input requires geometry processing algorithms. Most of these algorithms use polygonal meshes to represent shapes [11]. Polygonal meshes were created initially for researchers as a representation of real-world shapes that they can both interpret and manipulate [14, 52, 78]. However, developing algorithms that automatically manipulate meshes is often difficult since they involve discretizing the surface. For example, changing the topology of a mesh, such as turning a sphere into a torus, will break such discretization and require additional repair procedures such as remeshing [2, 3]. Given that shape editing is increasingly performed by algorithms, it seems worthwhile to search for a shape representation that is more amenable to automatic geometry processing.

An alternative to the polygonal mesh is an implicit representation, in which the surface is represented by a level set of a field: $\{x | f(x) = c\}$ [22, 39, 63]. Since it is easy to change shape topology using implicit representations, people have applied them for geometry processing tasks such as shape merging [8, 51, 54]. These works store implicit fields using voxels or octrees, which introduce memory-intensive spatial discretization. Recent research addresses this by using continuous neural networks to represent implicit fields [19, 45, 46, 56]. These *neural fields* possess several advantages in addition to the merits inherited from implicit representation: they are compact to store [23, 45] and can produce high-quality continuous surfaces at arbitrary resolutions [45, 56]. The community has achieved compelling results using neural fields in a variety of applications [47, 64, 73, 84]. Notwithstanding these results, it is still unclear whether shape editing tasks challenging for implicit fields (e.g., deformation and filtering) can be performed with neural fields. In this paper, we ask: *can geometry processing be done entirely using neural fields?*

---

*Email: gy46@cornell.edu. This work is done while Guandao was interning at Intel Labs.

35th Conference on Neural Information Processing Systems (NeurIPS 2021).

Geometry processing tasks are challenging because they are typically under-constrained: user input is very sparse. Therefore, we need to leverage priors that characterize how a natural surface behaves. Such surface priors are usually instantiated by minimizing energy functionals that describe the physical properties of surfaces using operators from differential geometry. For example, one can encourage surfaces to be smooth by minimizing the curvature, which can be measured by the Laplace-Beltrami operator. Such operators, however, require a parameterization of the surface, which is not available for level sets of a neural field. Mesh algorithms usually approximate these geometric operators using the geodesic neighbors of the surface point. This allows the algorithms to focus on intrinsic geometric properties that are independent of shape parameterization. However, it is not easy to obtain geodesic neighbors in a neural field since the surface of interest is encoded implicitly by a set of points that evaluate the field function to the same value.

We posit that computing these geometric operators in neural fields requires a fundamentally different approach. The idea is to approximate the local surface of the level set using the derivatives of the underlying field. We can evaluate intrinsic geometry properties of the level set, such as curvature, using only the field derivatives [39, 54]. This allows us to develop loss functions that describe surface priors such as elasticity or rigidity. To achieve this, we use the fact that neural fields are designed to be infinitely differentiable [66, 74]. The infinite differentiability of neural fields makes it possible to optimize loss functions that involve higher-order derivatives using gradient descent methods. Thus, unlike mesh-based geometry processing algorithms that approximate these objectives using surface discretizations, we can directly optimize in terms of the derivatives of the field.

Our formulation provides a proof of concept that geometry processing can be done entirely using neural fields. In particular, we focus on two tasks: shape filtering (e.g., smoothing and sharpening) and topology-preserving deformations. These two tasks are not only essential for downstream applications but also bring out the known challenges associated with implicit representations. We first tackle shape sharpening and smoothing and show how these can be done by optimizing a loss function based on geometric properties computed via the neural field's derivatives. To achieve shape deformation, we propose to warp the neural field using a deformation vector field modeled by an invertible neural network [7]. With this invertible deformation field, we derive a training objective that models the implicitly represented surfaces as elastic shells.

We compare our method with well-established mesh-based baselines [70, 76]. Our method can match the quality of the mesh-based counterparts without discretizing the surface. We hope that our work can inspire future generations of geometry processing algorithms using neural fields.

## 2    Related Work

This paper builds on two bodies of work: polygon mesh geometry processing and neural field representations. We will focus on prior work on two geometry processing tasks: shape filtering and topology-preserving deformation. Please refer to Botsch et al. [11] for further reading.

**Mesh smoothing (and sharpening).**    The goal of mesh smoothing is to remove high-frequency noise and produce a smooth surface. Noise removal can be achieved by Fourier analysis [76, 79] or by modeling surface motion with a diffusion equation [26]. Smoothness can be promoted by minimizing energy functionals inspired by differential operators [50, 72, 82]. Sharpening is the inverse of smoothing. In this paper, we adopt the goal of smoothing from prior work [20, 76] and design a corresponding differentiable objective for neural fields.

**Implicit field smoothing.**    Implicit representations can be smoothed by evolving the level set according to its curvature normal [49, 54]. Many network architectures, such as ReLU MLP, bias toward smoothness when modeling an implicit field [61, 66, 74]. Regularization objectives and initialization schemes have been proposed to obtain smooth level sets from scanned surface points [5, 31]. These works mostly use smoothness as a prior when training neural fields. Our work formulates an objective that allows smoothing or sharpening a neural field as desired.

**Mesh deformation.**    Mesh deformation is done by either deforming the shape (i.e. vertices) [9, 69] or the space [6, 36, 41, 48, 65]. These algorithms usually draw inspiration from physics and differential geometry to develop energy functions that encourage natural deformation [10, 15, 24, 70, 71, 77]. They also linearize such objectives so that they can be solved efficiently using linear

solvers [9]. Recent research also applies deep learning to optimize for non-linear deformation losses [32, 34, 38, 80, 81, 83]. Deforming a mesh, however, requires estimating the deformation objectives with spatial discretization and maintaining vertex connectivity in a way that preserves topology [42]. We deform neural fields with a continuous invertible field to circumvent these issues.

**Implicit field deformation.** Prior works have studied the deformation of implicit fields for tasks like physics simulation [39, 51, 54, 55]. These works use voxels or octrees to represent implicit fields, while our paper advocates for using neural fields for deformation. There is some past work on deforming neural fields to match a target shape [12, 18, 37, 53, 58] or image [43, 57, 60] without any correspondences between the two. In contrast, we perform such deformation with very few localized user-provided correspondences, but without any other information about the target shape. Our setting is very convenient for artists, but it requires the algorithm to provide a strong surface prior due to the lack of a densely specified target state. Other prior works on editing neural fields are *trained* on a dataset of shapes from a particular object category [25, 29, 30, 33, 85]. Since surface priors are specified implicitly by the dataset, it is unclear how to apply these methods to out-of-distribution shapes. In contrast, our paper requires no training dataset and instead enforces priors inspired by more general physical properties such as smoothness and elasticity. Remelli et al. [62] developed a way to differentiate iso-surface extraction, which allows deforming neural fields using objectives carried over from mesh-based algorithms. Our method avoids iso-surface extraction entirely and formulates differentiable deformation objectives directly on the implicit field.

## 3 Computing Surface Properties of Neural Fields

In this section, we will discuss how to compute surface properties using neural fields and establish notation for the following sections. Let $\partial\Omega$ be a surface enclosing the region $\Omega$. A signed distance field (SDF) for surface $\partial\Omega$ is defined as $d(\mathbf{x}) = s(\mathbf{x}) \min_{\mathbf{y} \in \partial\Omega} \|\mathbf{x} - \mathbf{y}\|$. The sign function $s(\mathbf{x})$ evaluates to $-1$ if $\mathbf{x} \in \Omega$ and 1 otherwise. We will assume that a neural field approximate an SDF whose zero-isosurface represents the surface of interest. For a neural field $f$, we denote its zero-isosurface using $\mathcal{M}_f$. This section summarizes how to compute surface normals and curvatures from neural fields. This is well-understood and we summarize this here for convenience [39, 54, 59].

**Surface normal.** The surface normal is the vector perpendicular to the tangent plane and describes the local orientation of the surface. For an SDF, the outward oriented surface normal of its level set is the field gradient: $\mathbf{n}_d(\mathbf{x}) = \nabla_{\mathbf{x}} d(\mathbf{x})$. It can be shown that the SDF's gradient norm is always one: $\|\nabla_{\mathbf{x}} d(\mathbf{x})\| = 1$. This property can be used to ensure that the field remains a valid SDF throughout any manipulation [4, 31]. The closest point in $\partial\Omega$ from $\mathbf{x}$ can be found by $\mathbf{x} - d(\mathbf{x})\mathbf{n}_d(\mathbf{x})$. This property can be used to sample points from the isosurface without creating a mesh [5, 13]. Finally, since the tangent plane is perpendicular to the surface normal, we can project vectors onto the tangent plane by subtracting their projection onto the normal using the following projection matrix $\mathbf{x}$: $\mathbf{P}_d(\mathbf{x}) = \mathbf{I} - \mathbf{n}_d(\mathbf{x})\mathbf{n}_d(\mathbf{x})^T$. This projection matrix allows us to characterize the tangent plane. This matrix is important for measuring how much a given deformation stretches the tangent direction [35].

**Curvature.** Intuitively, curvature describes how much a surface deviates from a plane. This can be captured by the total derivative of the surface normal, which is also known as the shape operator: $\mathcal{S}_D(\mathbf{x}) = \mathcal{D}\mathbf{n}_d(\mathbf{x})$. For an SDF, the normal itself is given by the derivative of the field, thus the shape operator for an SDF is the Hessian of the field function: $H_d(\mathbf{x}) = \mathcal{D}^2 d(\mathbf{x})$. The shape operator can be used to capture many different notions of curvature, including mean curvature (half the trace of $\mathcal{S}_D(\mathbf{x})$, denoted by $\bar{\kappa}$), Gaussian curvature (the determinant of $\mathcal{S}_D(\mathbf{x})$), and principal curvatures (the eigenvalues of $\mathcal{S}_D(\mathbf{x})$). We will use these to define objectives that smooth or sharpen the surface.

## 4 Shape Smoothing and Sharpening

In this section, we will show that shape smoothing and sharpening can be done directly on neural fields without producing meshes. In our setup, the input shape is represented as the zero-isosurface $\mathcal{M}_F$ of the neural field $F$. The algorithm needs to output neural field $G_\theta$ whose zero-isosurface $\mathcal{M}_{G_\theta}$ satisfies two goals. First, $\mathcal{M}_{G_\theta}$ should preserve the global structure of $\mathcal{M}_F$. Second, the surface $\mathcal{M}_{G_\theta}$ should make desired changes to match the curvature of shape $\mathcal{M}_F$.

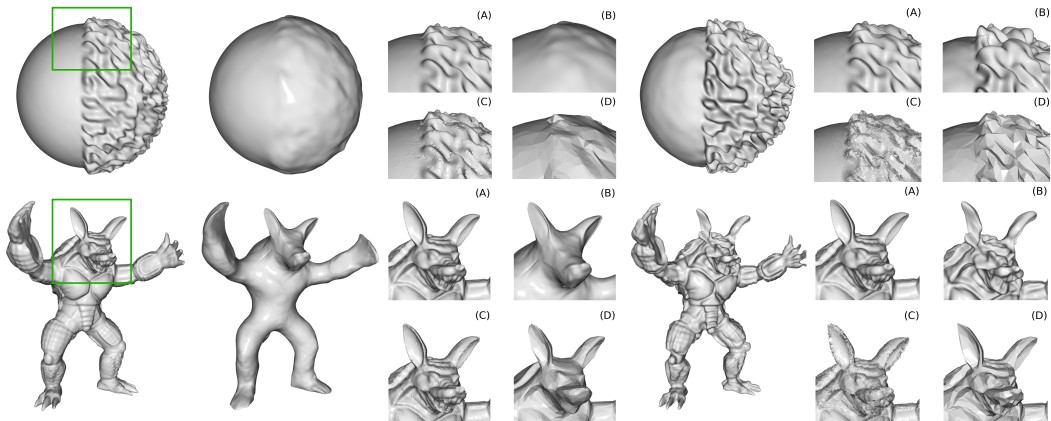

Figure 2: Smoothing and sharpening results. (A) zoomed-in input; (B) our method; (C) baseline without remeshing; (D) baseline with remeshing. We can see that the baseline without remeshing finds it difficult to smooth or sharpen the mesh extracted from neural fields. Our method is able to achieve good results without discretizing the surface.

To achieve these goals, we first instantiate $G_\theta$ using the same architecture and parameter values as $F$. Then we optimize $G_\theta$ using the following objective:

$$\mathcal{L}(\theta) = \int_{\mathbf{x} \in U} |G_\theta(\mathbf{x}) - F(\mathbf{x})|^2 + \lambda_g \left( \|\nabla_{\mathbf{x}} G_\theta(\mathbf{x})\| - 1 \right)^2 d\mathbf{x} + \int_{\mathbf{x} \in V_\tau} \lambda_k \left( \kappa_{G_\theta}(\mathbf{x}) - \beta \kappa_F(\mathbf{x}) \right)^2 d\mathbf{x}.$$

$$(1)$$

The first integral encourages the network $G_\theta$ to preserve the original shape. The second term regularizes $G_\theta$ to remain a valid SDF by enforcing that the norm of the gradient is 1 (i.e., the Eikonal constraint [4, 31]). The integration is over the region $U$ where the original neural field $F$ is supervised over. The final term aims to smooth or sharpen the surface by increasing or decreasing the curvature $\kappa$. Setting $\beta < 1$ will decrease the curvature of the output surface $\kappa_{G_\theta}$, resulting in a smoother shape. $\beta > 1$ will lead to surfaces with higher curvature details and will thus sharpen shapes. In this paper, we will use mean curvature since it's easy to compute with neural fields: $\kappa_f(\mathbf{x}) = \operatorname{tr} \mathcal{D}(\mathbf{n}_f(\mathbf{x}))$.

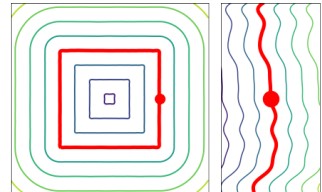

Figure 1: L: SIREN learned to fit the SDF of a square; R: Zooming in on the surface.

One challenge is that the computation of curvature can be very noisy for neural fields using periodic activations. Figure 4 shows how an ostensibly smooth isosurface learned through SIREN [66] is actually quite rough when zoomed in. The curvature evaluated on such a rough surface can be too noisy to be used for training. To alleviate this issue, we only compute the curvature regularization in areas where the curvature of the level set is less than a certain threshold $\tau$. Formally, we define this area as $V_\tau = \{\mathbf{x} \in U \mid \max(|\kappa_{G_\theta}|, |\kappa_F|) < \tau\}$. We use rejection sampling to sample points from $V_\tau$ when computing the loss during training.

**Results.** We follow prior works [20, 76] to use Armadillo [40] and a sphere with one half of it corrupted by Gaussian noise. We compare our algorithm to two baselines. The first baseline directly applies the smoothing algorithm developed by Taubin [76] to the mesh extracted from the neural field using Marching cube [44]. The second baseline applies the filtering algorithms on meshes simplified by quadratic decimation [27]. The input neural fields are created following the procedure of Park et al. [56]. The results are shown in Figure 2. The first baseline fails to smooth or sharpen the surface appropriately. It only modifies the surface with high-frequency noise. While the second baseline can filter the surface correctly, it introduces discretization artifacts due to the decimation process. This suggests it is challenging to filter the neural field surface with mesh processing algorithms since these algorithms are sensitive to the quality of surface discretization. On the other hand, our algorithm can produce good filtering results without the need to maintain a good discretization of the surface.

# 5 Deformation

To deform a shape, the user will choose a set of deformation handles $\{\mathbf{h}_i\}_{i=1}^n$. For each handles $\mathbf{h}_i$, the user will specify a target location $\mathbf{t}_i$ describing where the handle will be dragged to. Users can set $\mathbf{t}_i = \mathbf{h}_i$ to enforce a part of the surface to be unchanged. Given this input, there are two goals for the shape deformation algorithm: 1) satisfying user inputs and 2) ensuring that the deformation resembles the natural behavior of real objects. The input shape will be represented by the zero-isosurface of a neural field $F$ in our setting. The algorithm must therefore output a neural field $G_\theta$ whose zero-isosurface represents the deformed shape that satisfies the aforementioned two goals.

In this setting, deformation can be formulated as solving a constrained optimization problem:

$$\min_\theta \mathcal{L}_{dfm}(G_\theta, F) \quad \text{s.t. } \mathcal{L}_{const}(G_\theta, \mathbf{t}_i, \mathbf{h}_i) = 0, 1 \le i \le n, \tag{2}$$

where the objective $\mathcal{L}_{dfm}(G_\theta, F)$ measures how natural the deformation is. The constraints $\mathcal{L}_{const}(G_\theta, \mathbf{t}_i, \mathbf{h}_i) = 0$ ensure that the result deformation matches user input.

One way to ensure a natural-looking deformation is to assume that the surface behaves like a thin elastic shell, which is resistant to stretching and bending [1, 70]. Traditional mesh-based algorithms model elasticity by minimizing the thin shell energy that penalizes stretching or bending happened to any local patch. Naively adapting such training objectives from mesh-based algorithms to neural fields is challenging because these algorithms assume dense correspondences between two surfaces and efficient access to geodesic neighbors.

In this section, we will address these challenges by following strategy. To measure how much bending or stretching happened to the whole surface, we will first divide the surface into infinitesimal patches and then sum the amount of bending or stretching that happened to each patch. To achieve that, we first develop a sampling schema to sample these infinitesimal patches uniformly from the input shapes (Sec 5.1). Then we warp the input shape with an invertible neural network to achieve correspondences between the input and output surfaces (Sec 5.2). Finally, we derive ways that measure the amount of bending and stretching between two infinitesimal surface patches (Sec 5.3).

## 5.1 Sampling

Our first step is to sample infinitesimal surface patches uniformly to the surface area from the zero-isosurface of the neural field. Each infinitesimal surface patch can be represented by a surface point and its local derivatives. With this said, the problem amounts to sampling points from the zero-isosurface. Formally, we are given a neural field $F$ that approximates the signed distance field of the input shape. The goal is to find a way to sample an arbitrary number of points on its zero-isosurface $\mathcal{M}_F$. One way to sample points from $\mathcal{M}_F$ is to run Langevin dynamics

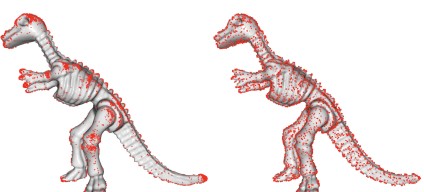

Figure 3: Sampling from dino. L: no rejection; R: ours (with rejection).

using the gradient of the field $F$ [5, 13]: $\mathbf{x}_{t+1} = \tilde{\mathbf{x}}_t - F(\tilde{\mathbf{x}}_t)n_F(\tilde{\mathbf{x}}_t)$, $\tilde{\mathbf{x}}_t \sim \mathcal{N}(\mathbf{x}_t, \sigma\mathbf{I})$. $\mathbf{x}_0$ are initialized by sampling uniformly within $[-1, 1]^3$. Here we also leverage an SDF property: the closest surface point to $\mathbf{x}$ can be computed by $\mathbf{x} - \mathbf{n}(\mathbf{x})F(\mathbf{x})$ (Sec 3). While this process produces points guaranteed to be on the zero-isosurface, the mixing time of the procedure is very long without appropriate coarse-to-fine annealing [68]. Running this process for limited iterations results in point samples that are concentrating near the high curvature area, as shown on the left side of Figure 5.1.

To alleviate this issue, we instantiate the sampling procedure with $\mathbf{x}_0$ sampled from a rough shape. This can be done by first sampling uniformly from the bounded space and rejecting points that are too far away from the isosurface: $\mathbf{x}_0 \sim \{\mathbf{x} \in U(-1, 1)|F(\mathbf{x}) < \tau_s\}$. We found that $\tau_s$ can be set to a relatively large number (e.g., about $0.1$). This prevents high rejection rates without sacrificing sampling quality. Figure 5.1 shows that our method creates uniform samples.

We quantitatively evaluate the effectiveness of the sampling scheme. Specifically, we compare our sampling scheme with directly applying Langevin dynamics [13] (*LD* in Table 1). To do that, we compute the Chamfer Distance (CD) and Earth Mover Distance (EMD) between points sampled by the valuated algorithms and points sampled uniformly from the extracted mesh. For both metrics, lower values indicate better performance Since larger surfaces can lead to larger CD and EMD values,

we normalize the metrics by the values obtained by comparing two sets of uniform samples from the mesh (i.e., CDr and EMDr) The results show that initializing $\mathbf{x}_0$ by rejection sampling significantly improves both CD and EMD compared to naively applying Langevin dynamics, at a minor cost in sampling time.

## 5.2 Invertible Deformation Field

After we obtained surface points samples from zero isosurfaces of the input neural fields, the next step is to deform these surface points to the output shapes. One way to deform neural fields is to warp the coordinate space of the input network using a deformation field predicted by a neural network $D_\theta$ [25, 43, 57]. Under this framework, the output network can be defined as $G_\theta(\mathbf{x}) = F(D_\theta(\mathbf{x}))$. Then the constraint $\mathcal{L}_{const}(G_\theta, \mathbf{t}, \mathbf{h})$ can be defined as $\mathcal{L}_{const}(G_\theta, \mathbf{t}, \mathbf{h}) = \|D_\theta(\mathbf{t}) - \mathbf{h}\|$.

Table 1: Our sampling method achieves better results than the baseline.

| Metrics | Dino | | Armadillo | |
|---|---|---|---|---|
| | LD | Ours | LD | Ours |
| CDr ($\downarrow$) | 1.54 | **1.04** | 1.36 | **1.02** |
| EMDr ($\downarrow$) | 3.38 | **1.15** | 3.30 | **1.08** |
| Time | **0.15** | 0.21 | **0.12** | 0.18 |

Ideally, we would like the deformation $D_\theta$ to be continuous and invertible. This will allow us to create well-behaved one-to-one correspondences between points on the deform shape (i.e., $\mathbf{x}$) and points on the input shape (i.e., $D_\theta(\mathbf{x})$). To achieve this, we make $D_\theta$ to be an invertible network composed of a sequence of invertible residual blocks [7].

**Invertible residual block.** Prior works [7, 17] have shown that a sufficient condition for the residual block $f(x) = x + g(x)$ to be invertible is that the Lipschitz constant of function $g$ is less than 1. The architecture of the network $g(x)$ is usually composed of spectral normalized linear layers and Lipschitz continuous nonlinearities such as ELU [21]. The inverse of such residual block can be computed by finding the fixed point of function $y \mapsto y - g(x)$ [7]. Intuitively, deforming using one invertible residual block amounts to moving point $x$ with deformation vector $g(x)$.

**Lipschitz continuous positional encoding.** Applying such architecture directly without modification fails to produce deformations with many different local rotations. Recent research suggests that periodic functions are essential for coordinate MLPs to predict complex signals [47, 66, 74]. Periodic functions are usually used as positional encoding or activations of the form $\sin(ax + b)$, where $a$ controls the frequency of the activation. To use such periodic function as part of the invertible residual block without restricting the frequency, we normalize the output of the periodic function $\sin(ax + b)|a|^{-1}$, bringing it is Lipschitz constant below 1. Formally, the positional encoding with normalized Lipschitz constant applied to one dimension is

$$\gamma_i(\mathbf{x}) = \frac{1}{\sqrt{2L+1}} \left( x_i, \frac{\cos(2^0 \pi \mathbf{x}_i)}{2^0 \pi}, \frac{\sin(2^0 \pi \mathbf{x}_i)}{2^0 \pi}, \ldots, \frac{\cos(2^L \pi \mathbf{x}_i)}{2^L \pi}, \frac{\sin(2^L \pi \mathbf{x}_i)}{2^L \pi} \right). \quad (3)$$

We will apply such encoding to each of the dimensions of the input coordinate for the body of the invertible residual block: $R_\theta(\mathbf{x}) = \mathbf{x} + g_\theta \left( \frac{1}{\sqrt{d}} [\gamma_1(\mathbf{x}), \ldots, \gamma_d(\mathbf{x})] \right).$

**Ablation.** We conduct ablation studies on two architecture choices: invertibility and positional encoding. In this experiment, we deform a neural field representing the SDF of a 2D rectangle. We optimize each ablation case to satisfies the user specified constraints with following loss: $\frac{1}{n} \sum_{i=1}^{n} \|D_\theta(\mathbf{h}_i) - \mathbf{t}_i\|^2$. The results are shown in Figure 4. If we replace the invertible architecture with SIREN as done in Deng et al. [25], the deformation field will tend to break topology (*No inverse*). Removing the positional encoding will fail to create a complex field (*No PE*). Our architecture can create a reasonable guess (*No loss*). This shows that our network architecture prioritizes natural deformation that preserves topology.

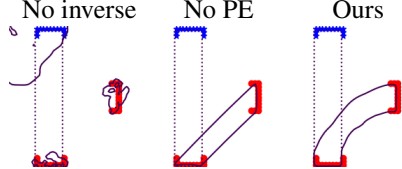

Figure 4: Architecture ablation on 2D SDF deformation. Blue points are handles; Red points are targets.

## 5.3 Implicit Thin Shell Loss

At this point, we have obtained the correspondences $\mathbf{y} = D_\theta(\mathbf{x})$, with $\mathbf{x}$ in the output (or deformed) space and $\mathbf{y}$ in the input space. Our next step is to design $\mathcal{L}_{dfm}$ that compares these corresponding patches to ensure natural deformation. Recall that one way to ensure a natural-looking deformation is to minimize the amount the resistance to bending or stretching that happened during the deformation [1, 70]. We will develop novel loss functions for measuring stretching and bending. Please refer to the supplement for the connection between our loss and the thin shell energy.

### 5.3.1 Stretching Loss

Stretching can be captured by the change of dot product in tangent space. Intuitively, a local surface patch is stretched when the lengths of some tangent vectors change. The change of tangent vectors' norm can be measured by the tangent dot-product. With this said, the amount of stretch can be measured by the change of tangent dot product.

Note that we do not have access to analytical surface parameterization, so we need to express the tangent dot product with neural fields and its derivative. We can describe the tangent dot product to the spatial dot product using the projection matrix. The tangent vector of $\mathbf{x}$ can be parameterized by projecting a vector $\mathbf{v}$ to the tangent space. To achieve that, we need to multiply $\mathbf{v}$ with the projection matrix $\mathbf{P}_{G_\theta}(\mathbf{x}) = \mathbf{I} - \mathbf{n}_{G_\theta}(\mathbf{x})\mathbf{n}_{G_\theta}(\mathbf{x})^T$, where $\mathbf{n}_{G_\theta}(\mathbf{x})$ is the surface normal of point $\mathbf{x}$. $\mathbf{P}_{G_\theta}(\mathbf{x})\mathbf{v}$ is a tangent vector in the tangent plane of $\mathbf{x}$.

Now we are ready to compute the change of tangent dot-product. Let $\mathbf{t}_i$ and $\mathbf{t}_j$ be two arbitrary tangent vectors near point $\mathbf{x}$ at the deformed shape. Further assume that these vectors can be parameterized as $\mathbf{t}_i = \mathbf{P}_{G_\theta}(\mathbf{x})\mathbf{v}_1$ and $\mathbf{t}_j = \mathbf{P}_{G_\theta}(\mathbf{x})\mathbf{v}_2$. These tangent vector will be transformed by $D_\theta$ into $\mathbf{t}_i' = \mathbf{J}_{D_\theta}(\mathbf{x})\mathbf{t}_i$ and $\mathbf{t}_j' = \mathbf{J}_{D_\theta}(\mathbf{x})\mathbf{t}_j$. These are tangent vectors at point $\mathbf{y}$ at the input shape. The change of tangent dot-product with respect to these two vectors can be computed as follows:

$$\left| \mathbf{t}_1^T \mathbf{t}_2 - \mathbf{t}_1'^T \mathbf{t}_2' \right| = \left| \mathbf{v}_1^T \mathbf{P}_{G_\theta}(\mathbf{x})^T \left( \mathbf{I} - \mathbf{J}_{D_\theta}(\mathbf{x})^T \mathbf{J}_{D_\theta}(\mathbf{x}) \right) \mathbf{P}_{G_\theta}(\mathbf{x})\mathbf{v}_2 \right|. \tag{4}$$

To minimize the stretch, we need to enforce that dot-product stays the same for all tangent vectors. This can be done by minimize the matrix norm of $\mathbf{P}_{G_\theta}^T \left( \mathbf{I} - \mathbf{J}_{D_\theta}^T \mathbf{J}_{D_\theta} \right) \mathbf{P}_{G_\theta}$. Here, we drop the function arguments $\mathbf{x}$ for the matrix for notation clarity. With these, we define the the stretch loss as:

$$\mathcal{L}_s(G_\theta) = \int_{\mathbf{x} \in \mathcal{M}_{G_\theta}} \left\| \mathbf{P}_{G_\theta}^T \left( \mathbf{I} - \mathbf{J}_{D_\theta}^T \mathbf{J}_{D_\theta} \right) \mathbf{P}_{G_\theta} \right\|_F^2 d\mathbf{x}. \tag{5}$$

### 5.3.2 Bending Loss

Bending can be characterized by the change of surface curvature (e.g., making the surface more or less curved). Intuitively, curvature can be described as the change of tangent dot product along the surface normal direction [16, 59]. Let $\mathbf{t}_1 = \mathbf{P}_{G_\theta}(\mathbf{x})\mathbf{v}_1$ and $\mathbf{t}_2 = \mathbf{P}_{G_\theta}(\mathbf{x})\mathbf{v}_2$ be two tangent vectors at $\mathbf{x}$. Consider the tangent dot product of the surface family $\mathbf{x} + t\mathbf{n}(\mathbf{x})$. Note that this set of surface corresponding to the level sets $\{\mathbf{p} | G_\theta(\mathbf{p}) = t\}$ if $G_\theta$ is approximating an SDF [54]. Then the change of dot product along the surface normal direction can be given by Hessian and directional derivative: $\frac{d}{dt} \mathbf{t}_1^T \mathbf{t}_2 |_{t=0} = \mathbf{t}_1^T H_{G_\theta}(\mathbf{x})\mathbf{t}_2$. Intuitively, the larger this value is, the faster the surface changes when going along the surface normal direction, which means the surface has larger curvature.

Measuring bending amounts to measure the change of tangent dot-product derivative along the surface normal direction. Similar to the previous section, assume we have tangent vectors $\mathbf{t}_i = \mathbf{P}_{G_\theta}(\mathbf{x})\mathbf{v}_i$ and $\mathbf{t}_j = \mathbf{P}_{G_\theta}(\mathbf{x})\mathbf{v}_j$. These tangent vectors are are transformed by the Jacobian of the deformation field $\mathbf{J}_{D_\theta}(\mathbf{x})$ to $\mathbf{t}_i' = \mathbf{J}_{D_\theta}(\mathbf{x})\mathbf{t}_i$ and $\mathbf{t}_j' = \mathbf{J}_{D_\theta}(\mathbf{x})\mathbf{t}_j$. Since $\mathbf{t}_i'$ and $\mathbf{t}_j'$ are tangent vectors at point $\mathbf{y}$ on the surface $\mathcal{M}_F$, the derivative of tangent dot-product is given by the hessian matrix of field $F$: $\frac{d}{dt} \mathbf{t}_1'^T \mathbf{t}_2' |_{t=0} = \mathbf{t}_1'^T H_F(D_\theta(\mathbf{x}))\mathbf{t}_2' = \mathbf{t}_1^T \mathbf{J}_{D_\theta}^T(\mathbf{x}) H_F(D_\theta(\mathbf{x}))\mathbf{J}_{D_\theta}(\mathbf{x})\mathbf{t}_2$. We will drop the function argument to $\mathbf{J}_{D_\theta}$, $H_{G_\theta}$, and $\mathbf{P}_{G_\theta}$ for notation clarity. The change of tangent dot-product derivative is:

$$\left| \frac{d}{dt} \left( \mathbf{t}_1^T \mathbf{t}_2 - \mathbf{t}_1'^T \mathbf{t}_2' \right)_{t=0} \right| = \left| \mathbf{v}_1^T \mathbf{P}_{G_\theta}^T \left( H_{G_\theta} - \mathbf{J}_{D_\theta}^T H_F(D_\theta(\mathbf{x}))\mathbf{J}_{D_\theta} \right) \mathbf{P}_{G_\theta}\mathbf{v}_2 \right|. \tag{6}$$

If there is almost no bending happened between the infinitesimal patches around $\mathbf{x}$ and $\mathbf{y}$, then the quantity $\left| \frac{d}{dt} \left( \mathbf{t}_1^T \mathbf{t}_2 - \mathbf{t}_1'^T \mathbf{t}_2' \right)_{t=0} \right|$ should stay close to 0 for all pairs of tangent vectors. We will

quantify this by the matrix norm of $\mathbf{P}_{G_\theta}^T \left(H_{G_\theta} - \mathbf{J}_{D_\theta}^T H_F(D_\theta(\mathbf{x}))\mathbf{J}_{D_\theta}\right)\mathbf{P}_{G_\theta}$. Finally, we arrive at our bending loss, which minimizes the norm of the matrix that modulate the change of tangent dot-product derivatives moving along the surface normal direction:

$$\mathcal{L}_b(G_\theta) = \int_{x \in \mathcal{M}_{G_\theta}} \left\|\mathbf{P}_{G_\theta}^T \left(H_{G_\theta} - \mathbf{J}_{D_\theta}^T H_F(D_\theta(\mathbf{x}))\mathbf{J}_{D_\theta}\right)\mathbf{P}_{G_\theta}\right\|_F^s d\mathbf{x}. \tag{7}$$

### 5.3.3 Computing $\mathcal{L}_s$ and $\mathcal{L}_b$

Computing $\mathcal{L}_s$ and $\mathcal{L}_b$ during training requires approximating a surface integral of the form $\int_{\mathbf{x} \in \mathcal{M}_{G_\theta}} \mathcal{L}(\mathbf{x}) d\mathbf{x}$ using Monte Carlo integration, which requires an efficient way to sample points from surface $\mathcal{M}_{G_\theta}$. While it is feasible to sample uniformly from an SDF, this is challenging to do with $G_\theta$ since there is no guarantee that $G_\theta$ remains a valid SDF during the course of training. Thanks to the invertibility of $D_\theta$, we can apply change of variable $\mathbf{x} = D_\theta^{-1}(\mathbf{y})$ to the integration, where $\mathbf{y}$ are points on the input surface $\mathcal{M}_F$:

$$\int_{\mathbf{x} \in \mathcal{M}_{G_\theta}} \mathcal{L}(\mathbf{x}) d\mathbf{x} = \int_{\mathbf{y} \in \mathcal{M}_F} \mathcal{L}(\mathbf{x}) \left|\det \left(\mathbf{J}_{D_\theta}(\mathbf{x})\mathbf{P}_{G_\theta}(\mathbf{x}) + \mathbf{n}_F(\mathbf{y})\mathbf{n}_{G_\theta}(\mathbf{x})^T\right)\right|^{-2} d\mathbf{y}. \tag{8}$$

$\mathbf{J}_{D_\theta}\mathbf{P}_{G_\theta} \in \mathbb{R}^{3 \times 3}$ maps tangent vector from $\mathcal{M}_{G_\theta}$ to $\mathcal{M}_F$. We apply the extension trick from Iglesias et al. [35] to create a matrix whose determinant equals the change of surface area. This is achieved by adding the surface normal component $\mathbf{n}_F\mathbf{n}_G^T$ and keeping it unchanged. With this change of variable, we now can compute the losses with point samples from the input field $F$. Since the input field $F$ approximates an SDF, we can use the method introduced in Section 5.1 to sample uniform points.

## 5.4 Optimization

Putting the losses together, we deform a neural field by solving the constrained optimization problem

$$\arg\min_\theta \lambda_s \mathcal{L}_s(G_\theta) + \lambda_b \mathcal{L}_b(G_\theta), \quad \text{s.t. } \forall 1 \leq i \leq n, \; \|D_\theta(\mathbf{t}_i) - \mathbf{h}_i\| = 0 \tag{9}$$

where $\lambda_s$, $\lambda_b$ are hyperparameters that determine the material properties. One way to solve this constrained optimization problem for a neural network is to make the constraints a soft loss adding to the objective function: $\mathcal{L}_{const}(D_\theta) = \frac{1}{n}\sum_{1=1}^n \|D_\theta(\mathbf{t}_i) - \mathbf{h}_i\|$. Our final objective is:

$$\mathcal{L}(\theta) = \mathcal{L}_s(G_\theta) + \lambda_b \mathcal{L}_b(G_\theta) + \lambda_c \mathcal{L}_c(D_\theta). \tag{10}$$

We set $\lambda_c$ to be a high value to enforce that the model satisfies user-specified input as much as possible. The user can tune $\lambda_b$ and $\lambda_s$ depending on the application scenario. We optimize $\mathcal{L}(\theta)$ with Adam optimizer to obtain the output field $G_\theta$. Hyperparameters are provided in the supplement.

## 6 Deformation Results

In this section, we will demonstrate the results of our methods to shape deformation. The shapes for deformation are taken from Sorkine-Hornung and Alexa [70]. To create neural fields from these meshes, we follow the procedure of Park et al. [56] to compute ground-truth SDF for locations sampled within $[-1, 1]^3$. We then fit a SIREN [66] to the ground-truth SDF to generate our initial neural fields. Our main baseline is ARAP [70, 86]. The simplest way to use ARAP to edit neural fields is applying it on a mesh extracted from the input neural fields using marching cubes [44, 45]. We first present results comparing with ARAP applied to the extracted mesh. Then we will conduct an analysis to show our losses encourage the right behaviors.

**Comparing to ARAP baseline.** We follow ARAP [70] to create a set of basic shape deformation operations to evaluate our algorithm. In this setting, the user will first specify a set of handles used for manipulation (shown in Figure 5 in blue). In general, the user can apply three basic types of operation on a handle: 1) make it static (i.e., no deformation), 2) translate it, or 3) rotate it around a center. Usually, the first operation is used in combination with the latter two to produce useful deformation. We first show how our algorithm deforms simple objects (e.g., a cylinder and a rectangle bar) when the user rotates or translates one end while fixing the other. We then extend this set of operations to shapes with more detail (e.g., Cactus, Armadillo, and Dino). Finally, we test our algorithms when all three operations are specified together on a single shape. The results are shown in Figure 5. We can see that applying ARAP directly on the extracted mesh creates undesirable volume distortion. Our algorithm is able to produce deformation results that are natural while satisfying the user's intention.

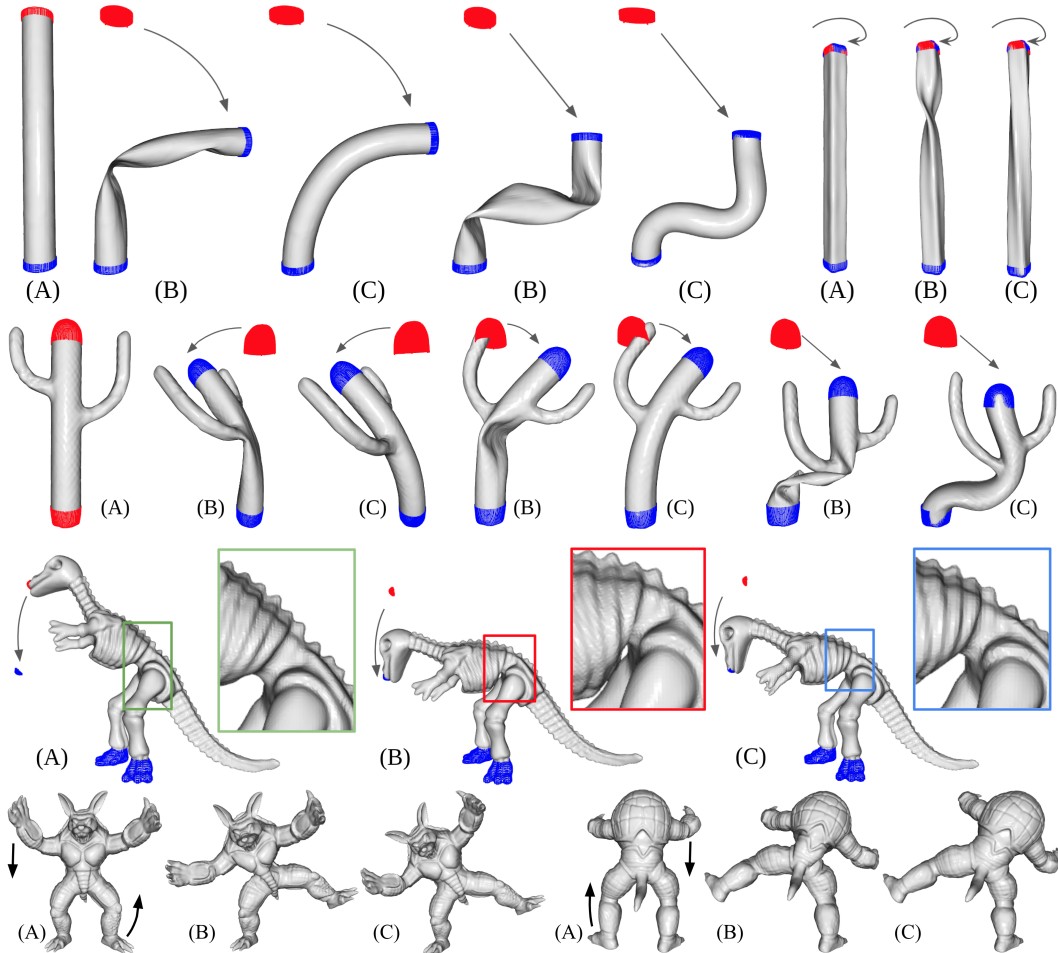

Figure 5: Deformation results. (A) Input shape. (B) Baseline. (C) Ours. Red points are user specified handles $\mathbf{h}_i$. Blue points denote user specified target points $\mathbf{t}_i$. First row: single rotation or translation on simple shapes. Second and third rows: single rotation or translation on more complex objects. Fourth row: multiple edits on complex objects.

**Comparing to different ARAP variants.** This artifact is partially due to the surface discretization made by the marching cubed algorithm doesn't agree with the assumptions made by ARAP [41]. To verify this hypothesis, we also include three additional baselines in Figure 6. First, we show the results of applying ARAP on the low-poly mesh used to create the input neural fields (*Original*). Second, we first apply Garland and Heckbert [28] to simplify the surface of the extracted mesh before applying ARAP to it (*Remeshed*). Finally, we also compare to the results of SR-ARAP [41], which adds smoothness regularization to ARAP to remove some unnatural distortion (*SR-ARAP*). The figure shows that ARAP works much better when the surface is discretized appropriately. Discretizing the surface, however, usually requires case-by-case manual adjustment to perform well. Our algorithm does not suffer from such issues as we optimize the thin shell objective directly without committing to a particular discretization of the surface.

**Loss analysis.** Here we provide an experiment to analyze the behavior of stretching loss and bending loss. Specifically, we compare models optimized for only the stretching loss $\mathcal{L}_s$, only the bending loss $\mathcal{L}_b$, and both losses together. Similar to Section 5.2, we deform a neural field that approximates the SDF of a 2D rectangle. The results are shown in Figure 7. Optimizing only for $\mathcal{L}_s$ produces a shape that best preserves surface area (or the perimeter in the 2D case), which explains the wiggling. Optimizing only for $\mathcal{L}_b$ produces a shape that tries to prevent wiggling of the surface, which changes the curvature of the surface. The user can choose appropriate $\lambda_s$ and $\lambda_b$ to produce deformation with resistance to both bending and stretching.

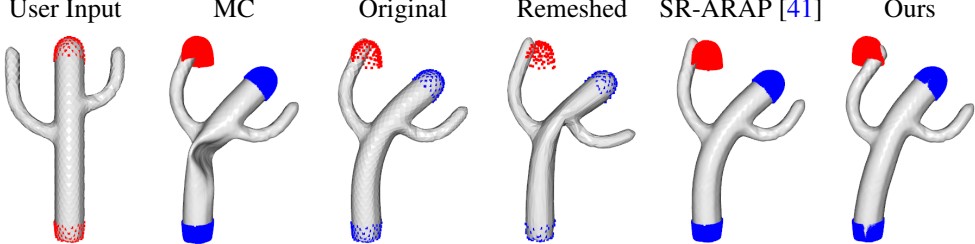

| User Input | MC | Original | Remeshed | SR-ARAP [41] | Ours |

Figure 6: Additional ARAP baselines. Directly applying ARAP to extracted mesh produces poor results (i.e., *MC*). While this can be alleviated by remshing (i.e. *Original* or *Remeshed*) or regularization (i.e. *SR-ARAP*), this shows mesh deformation algorithms can be sensitive to discretization quality. Our method do not discretize the surface, thus circumvent such issue. Red points are handles; Blue points are targets.

## 7 Discussion, Limitations, and Future work

**Advantages of using neural fields for geometry processing.** The main strength of using neural fields for geometry processing is that one can disentangle the processing algorithm from the surface discretization. Users do not need to worry about discretization while manipulating a shape represented in neural fields. This might open the door to more automatic geometry processing pipelines. Additionally, neural fields are modular and can be combined with other neural networks. Neural fields can be easily incorporated into current deep learning pipelines. This makes them well-suited for data-driven applications.

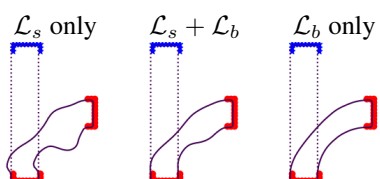

Figure 7: Analysis of stretching and bending losses. Blue points are handles; Red points are targets.

**Limitations and future directions.** The main limitation of our method is the slow optimization speed that hinders interactive editing. Typically, ARAP can solve each of such deformation problems within half an hour for large meshes, and it can achieve interactive speed for meshes with a small number of triangles. The Taubin smoothing can finish in a couple of seconds. Our deformation method right now requires a TitanX GPU with 12GB memory to train for 10 hours. Our smoothing and sharpening method takes about 10 minutes on the same GPU. The current algorithm does not ensure that the output field remains an SDF, which prevents performing multiple editing operations.

We believe many of these limitations can be addressed in future work. One can potentially use techniques like progressive training or meta-learning [67, 75] to improve the training speed. To allow consecutive editing, we can design regularization loss that enforces the output fields to approximate an SDF [4] or design sampling methods robust with noisy SDF. Other interesting directions include providing deeper theoretical analysis and applying neural fields in other geometry processing tasks.

**Societal impact.** Our work can lead to more efficient geometry processing algorithms. Such algorithms can empower artists to generate creative content. Potential negative impact includes misuse of geometry processing algorithms to create offensive content.

## 8 Conclusion

Our work provides a proof of concept that geometry processing can be done entirely with neural fields without discretizing the surface. We develop network architectures and training objectives for filtering and deforming shapes represented by neural fields and demonstrate the advantages of using neural fields for geometry processing. We hope that our work can inspire a new generation of geometry processing algorithms using neural fields.

**Acknowledgement.** Guandao's PhD was supported in part by a research gift from Magic Leap and a donation from NVIDIA. We want to thank Wenqi Xian, Professor Steve Marschner, and members of Intel Labs for providing insightful feedback for this project.

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
