# Geometry Processing with Neural Fields
## - Supplementary Materials -

**Guandao Yang** [*]
Cornell University

**Serge Belongie**
University of Copenhagen

**Bharath Hariharan**
Cornell University

**Vladlen Koltun**
Intel Labs

## Contents

## 1 Overview

This supplementary material contains theoretical derivations, implementation details, and additional results and discussions. We will review some useful properties of neural fields (Section 2). Then we will show the derivation of some important properties of the loss functions in Section 3. We also show the connection between thin shell energy and our losses in Section 3. Section 4 contains the derivation for the change of variable formula (i.e. Main paper equation (8)). We will provide the implementation details in Section 5). Some additional results are shown in Section 6 to understand our method's behavior better. Finally, we will provide a more detailed discussion of limitations and future works (Section 7).

## 2 Basic Properties

In this section, we will extend on some properties of the implicit fields. In particular, we will first show some properties of the projection matrix. Then we will show some properties of surface normal and tangent vectors. These properties will be used in later sections (i.e. Sec 3 and Sec 4).

---

[*]Email: gy46@cornell.edu. This work is done while Guandao is interning in Intel Labs.

35th Conference on Neural Information Processing Systems (NeurIPS 2021).

**Theorem 2.1.** (*Properties of surface normal and projections*) *For surface normal vector* $\mathbf{n}$, *we have the followings:*

    1. *The matrix* $\mathbf{P} = \mathbf{I} - \mathbf{nn}^T$ *projects vectors onto the tangent plane.*

    2. $\mathbf{P}^T = \mathbf{P} = \mathbf{PP}$.

*Proof.* For the first property, we want to show for all vector $\mathbf{v}$, $\mathbf{Pv}$ is perpendidular to $\mathbf{n}$:

$$
\begin{aligned}
(\mathbf{Pv})^T \mathbf{n} &= \mathbf{v}^T \left(I - \mathbf{nn}^T\right)^T \mathbf{n} \\
&= \mathbf{v}^T \mathbf{n} - \mathbf{v}^T \mathbf{n}^T \mathbf{nn} \\
&= \mathbf{v}^T \mathbf{n} - \mathbf{v}^T \mathbf{n} = 0
\end{aligned}
$$

We now need to show that $\mathbf{Pv}$ is the shortest point of $\mathbf{v}$ to the tangent plane. This can be seen as $\mathbf{v} - \mathbf{Pv} = \mathbf{nn}^T \mathbf{v}$ is a vector perpendicular to the tangent plane. The length of $\mathbf{v} - \mathbf{Pv}$ equals the projection of $\mathbf{v}$ onto the surface normal direction: $\mathbf{n}^T \mathbf{v}$.

Second properties can be seen by simple algebra:

$$
\begin{aligned}
\mathbf{PP} &= \left(\mathbf{I} - \mathbf{nn}^T\right)\left(\mathbf{I} - \mathbf{nn}^T\right) \\
&= \mathbf{I} - 2\mathbf{nn}^T + (\mathbf{nn}^T)^T(\mathbf{nn}^T) \\
&= \mathbf{I} - 2\mathbf{nn}^T + \mathbf{n}^T \mathbf{nnn}^T \\
&= \mathbf{I} - 2\mathbf{nn}^T + \mathbf{nn}^T \\
&= \mathbf{I} - \mathbf{nn}^T = \mathbf{P}
\end{aligned}
$$

and $\mathbf{P}^T = \mathbf{I} - \left(\mathbf{nn}^T\right)^T = \mathbf{I} - (\mathbf{n}^T)^T \mathbf{n}^T = \mathbf{I} - \mathbf{nn}^T = \mathbf{P}$.

$\square$

**Theorem 2.2.** (*Transformation in Tangent and Surface normal directions*) *Let* $F : \mathbb{R}^3 \to \mathbb{R}$ *be a signed distance field, and* $D : \mathbb{R}^3 \to \mathbb{R}^3$ *is an invertible deformation field. Define a new field* $G$ *by* $G(\mathbf{x}) = F(D(\mathbf{x}))$, *then the following will be true:*

    1. *Let* $\mathbf{y} = D(\mathbf{x})$, *then* $n_G(\mathbf{x}) = \frac{\mathbf{J}_D(\mathbf{x})^T n_F(\mathbf{y})}{\|\mathbf{J}_D(\mathbf{x})^T n_F(\mathbf{y})\|}$ *(thus* $n_F(\mathbf{y}) = \frac{\mathbf{J}_D(\mathbf{x})^{-T} n_G(\mathbf{x})}{\|\mathbf{J}_D(\mathbf{x})^{-T} n_G(\mathbf{x})\|}$*).*

    2. *Let* $\mathbf{u} \in \mathbb{R}^3$ *be a vector that* $\mathbf{u}^T n_G(\mathbf{x}) = 0$ *(i.e. in the tangent plane of* $G_\theta$ *at point* $\mathbf{x}$*), then* $\left(\mathbf{J}_D(\mathbf{x})\mathbf{u}\right)^T n_F(\mathbf{y}) = 0$ *(i.e. in the tangent plane of* $F$ *at point* $\mathbf{y}$*).*

    3. $\mathbf{P}_F(\mathbf{y})\mathbf{J}_D(\mathbf{x})\mathbf{P}_G(\mathbf{x}) = \mathbf{J}_D(\mathbf{x})\mathbf{P}_G(\mathbf{x})$.

*Proof.* By chain rule, we have $\nabla_x G(\mathbf{x}) = \left(\nabla_{\mathbf{y}} F(\mathbf{y})\right)^T \mathbf{J}_D(\mathbf{x}) = \mathbf{J}_D(\mathbf{x})^T n_F(\mathbf{y})$ since $F$ is a signed distance field (i.e. $n_F(\mathbf{y}) = \nabla_{\mathbf{y}} F(\mathbf{y})$). This implies:

$$
n_G(\mathbf{x}) = \|\nabla_{\mathbf{x}} G(\mathbf{x})\|^{-1} \mathbf{J}_D(\mathbf{x})^T n_F(\mathbf{y}) = \frac{\mathbf{J}_D(\mathbf{x})^T n_F(\mathbf{y})}{\|\mathbf{J}_D(\mathbf{x})^T n_F(\mathbf{y})\|}.
$$

With this we can prove the second part:

$$
\left(\mathbf{J}_D(\mathbf{x})\mathbf{u}\right)^T n_F(\mathbf{y}) = \mathbf{u}^T \left(\mathbf{J}_D(\mathbf{x})^T n_F(\mathbf{y})\right) = \mathbf{u}^T n_G(\mathbf{x}) \left\|\mathbf{J}_D(\mathbf{x})^T n_F(\mathbf{y})\right\| = 0.
$$

For the third property, we have:

$$\begin{aligned}
\mathbf{P}_F \mathbf{J}_D \mathbf{P}_G &= \left(\mathbf{I} - n_F n_F^T\right) \mathbf{J}_D \left(\mathbf{I} - n_G n_G^T\right) \\
&= \left(\mathbf{I} - n_F n_F^T\right) \left(\mathbf{J}_D - \mathbf{J}_D n_G n_G^T\right) \\
&= \mathbf{J}_D - \mathbf{J}_D n_G n_G^T - n_F n_F^T \mathbf{J}_D + n_F n_F^T \mathbf{J}_D n_G n_G^T \\
&= \mathbf{J}_D - \mathbf{J}_D n_G n_G^T - n_F \left(\mathbf{J}_D^T n_F\right)^T + n_F \left(\mathbf{J}_D^T n_F\right)^T n_G n_G^T \\
&= \mathbf{J}_D - \mathbf{J}_D n_G n_G^T - \frac{n_F n_G^T}{\|\mathbf{J}_D n_F\|} + \frac{n_F n_G^T n_G n_G^T}{\|\mathbf{J}_D n_F\|} \\
&= \mathbf{J}_D - \mathbf{J}_D n_G n_G^T - \frac{n_F n_G^T}{\|\mathbf{J}_D n_F\|} + \frac{n_F n_G^T}{\|\mathbf{J}_D n_F\|} \\
&= \mathbf{J}_D \left(\mathbf{I} - n_G n_G^T\right) = \mathbf{J}_D \mathbf{P}_G.
\end{aligned}$$

$\square$

# 3 Properties of Implicit Thin Shell Losses

In this section, we will show some properties of the stretching and bending loss. We first discuss the properties when stretching and bending loss reaches $0$. We then discuss the relationship between these two losses and the thin-shell energy.

## 3.1 Stretching and Bending Loss Reaches Zero

In particular, we want to show that when the stretching or bending loss reaches zero at a point, its counterpart that assumes a particular parameterization also reaches zero.

**Theorem 3.1.** *(**Stretch Loss**) Assume the iso-surface $\mathcal{M}_f$ of field $f$ is parameterized by $\mathbf{y}(u,v) \in \mathbb{R}^3$. Field $g$ is defined by $g = f(D(x))$ with an invertible function $D : \mathbb{R}^3 \to \mathbb{R}^3$. Let $\mathbf{x} = D^{-1}(\mathbf{y})$. Let $\mathbf{I}_f(\mathbf{y})$ and $\mathbf{I}_g(\mathbf{x})$ be the first fundamental form of $\mathcal{M}_f$ and $\mathcal{M}_g$ for point $\mathbf{y}$ and $\mathbf{x}$ respectively. If*

$$\left\| \mathbf{P}_g(\mathbf{x})^T \mathbf{P}_g(\mathbf{x}) - \mathbf{P}_g(\mathbf{x})^T \mathbf{J}_D(\mathbf{x})^T \mathbf{J}_D(\mathbf{x}) \mathbf{P}_g(\mathbf{x}) \right\|_F = 0, \tag{1}$$

*then we will have*

$$\left\| \mathbf{I}_g(\mathbf{x}) - \mathbf{I}_f(\mathbf{y}) \right\|_F = 0, \tag{2}$$

*where $\|\cdot\|_F$ denotes the Frobenius norm.*

*Proof.* Since $\mathbf{y} = D(\mathbf{x})$, so the surface $\mathcal{M}_g$ can be parameterized by $\mathbf{x}(u,v) = D^{-1}(\mathbf{y}(u,v))$. Note that $\mathbf{x}(u,v) \in \mathbb{R}^3$. Then by chain rule we have $\mathbf{x}_u = \mathbf{J}_{D^{-1}}(\mathbf{y})\mathbf{y}_u$ and $\mathbf{x}_v = \mathbf{J}_{D^{-1}}(\mathbf{y})\mathbf{y}_v$. Since $D$ is invertible, we have $\mathbf{y}_u = \mathbf{J}_D(\mathbf{x})\mathbf{x}_u$ and $\mathbf{y}_v = \mathbf{J}_D(\mathbf{x})\mathbf{x}_v$.

Now we compute the difference between two fundamental forms:

$$\begin{aligned}
\mathbf{I}_g(\mathbf{x}) - \mathbf{I}_f(\mathbf{y}) &= \begin{bmatrix} \mathbf{x}_u^T \mathbf{x}_u & \mathbf{x}_u^T \mathbf{x}_v \\ \mathbf{x}_v^T \mathbf{x}_u & \mathbf{x}_v^T \mathbf{x}_v \end{bmatrix} - \begin{bmatrix} \mathbf{y}_u^T \mathbf{y}_u & \mathbf{y}_u^T \mathbf{y}_v \\ \mathbf{y}_v^T \mathbf{y}_u & \mathbf{y}_v^T \mathbf{y}_v \end{bmatrix} \\
&= \begin{bmatrix} \mathbf{x}_u^T \mathbf{x}_u & \mathbf{x}_u^T \mathbf{x}_v \\ \mathbf{x}_v^T \mathbf{x}_u & \mathbf{x}_v^T \mathbf{x}_v \end{bmatrix} - \begin{bmatrix} \mathbf{x}_u^T \mathbf{J}_D^T \mathbf{J}_D \mathbf{x}_u & \mathbf{x}_u^T \mathbf{J}_D^T \mathbf{J}_D \mathbf{x}_v \\ \mathbf{x}_v^T \mathbf{J}_D^T \mathbf{J}_D \mathbf{x}_u & \mathbf{x}_v^T \mathbf{J}_D^T \mathbf{J}_D \mathbf{x}_v \end{bmatrix} \\
&= \mathbf{B}(\mathbf{x})^T \left(\mathbf{I} - \mathbf{J}_D^T \mathbf{J}_D\right) \mathbf{B}(\mathbf{x}),
\end{aligned}$$

where $\mathbf{B}(\mathbf{x}) = [\mathbf{x}_v \ \mathbf{x}_u]$ is a $3 \times 2$ matrix (Since $\mathbf{x}_u$ and $\mathbf{x}_v$ are all $\mathbb{R}^3$). We want to show that if Equation 1 holds, then for all vectors $\mathbf{w}_1 = [u_1 \ v_1] \in \mathbb{R}^2$ and $\mathbf{w}_2 = [u_2 \ v_2] \in \mathbb{R}^2$, we will have $\mathbf{w}_1^T \left(\mathbf{I}_g - \mathbf{I}_f\right) \mathbf{w}_2 = 0$.

Let $\mathbf{t}_1 = \mathbf{B}(\mathbf{x})\mathbf{w}_1 \in \mathbb{R}^3$ and $\mathbf{t}_2 = \mathbf{B}(\mathbf{x})\mathbf{w}_2$, which are two tangent vectors at point $\mathbf{x}$ on iso-surface $\mathcal{M}_g$. We want to show that $\mathbf{t}_1^T \left(\mathbf{I} - \mathbf{J}_D^T \mathbf{J}_D\right) \mathbf{t}_2 = 0$. One way to see that is to reparameterize the tangent vectors with

$$\mathbf{t}_{1,2} = \mathbf{P}_g(\mathbf{x}) \left(u_{1,2}\mathbf{x}_u + v_{1,2}\mathbf{x}_v\right) = \mathbf{P}_g(\mathbf{x})\mathbf{t}_{1,2},$$

which holds because $\mathbf{x}_u^T n_G(\mathbf{x}) = \mathbf{x}_v^T n_G(\mathbf{x}) = 0$ by definition (i.e. $n_G(\mathbf{x}) = \frac{\mathbf{x}_u \times \mathbf{x}_v}{\|\mathbf{x}_u \times \mathbf{x}_v\|}$). With the reparameterization, we have

$$
\begin{aligned}
\mathbf{w}_1^T \left(\mathbf{I}_g - \mathbf{I}_f\right) \mathbf{w}_2 &= \mathbf{t}_1^T \left(\mathbf{I} - \mathbf{J}_D^T \mathbf{J}_D\right) \mathbf{t}_2 \\
&= \mathbf{t}_1^T \mathbf{P}_g(\mathbf{x})^T \left(\mathbf{I} - \mathbf{J}_D(\mathbf{x})^T \mathbf{J}_D(\mathbf{x})\right) \mathbf{P}_g(\mathbf{x})\mathbf{t}_2 \\
&= \mathbf{t}_1^T \left(\mathbf{P}_g(\mathbf{x})^T \mathbf{P}_G(\mathbf{x}) - \mathbf{P}_g(\mathbf{x})^T \mathbf{J}_D(\mathbf{x})^T \mathbf{J}_D(\mathbf{x})\mathbf{P}_g(\mathbf{x})\right) \mathbf{t}_2 \\
&= 0,
\end{aligned}
$$

since $\left\|\mathbf{P}_g(\mathbf{x})^T \mathbf{P}_G(\mathbf{x}) - \mathbf{P}_g(\mathbf{x})^T \mathbf{J}_D(\mathbf{x})^T \mathbf{J}_D(\mathbf{x})\mathbf{P}_G(\mathbf{x})\right\|_F = 0$. $\qquad\square$

**Theorem 3.2.** *(**Bending Loss**)* *Assume the iso-surface $\mathcal{M}_f$ of field $f$ is parameterized by $\mathbf{y}(u, v) \in \mathbb{R}^3$. Field $g$ is defined by $g = f(D(x))$ with an invertible function $D : \mathbb{R}^3 \to \mathbb{R}^3$. Let $\mathbf{x} = D^{-1}(\mathbf{y})$. Let $\mathbf{II}_f(\mathbf{y})$ and $\mathbf{II}_g(\mathbf{x})$ be the first fundamental form of $\mathcal{M}_f$ and $\mathcal{M}_g$ for point $\mathbf{y}$ and $\mathbf{x}$ respectively. Let $\mathcal{S}_f(\mathbf{y})$ and $\mathcal{S}_g(\mathbf{x})$ be the shape operator of $\mathcal{M}_f$ and $\mathcal{M}_g$ for point $\mathbf{y}$ and $\mathbf{x}$ respectively. If*

$$
\left\|\mathbf{P}_g^T \left(\mathcal{D}(n_g) - \mathbf{J}_D^T \mathcal{D}(n_f)\mathbf{J}_D\right) \mathbf{P}_g\right\|_F = 0, \tag{3}
$$

*then we will have*

$$
\left\|\mathbf{II}_g(\mathbf{x}) - \mathbf{II}_f(\mathbf{y})\right\|_F = 0, \tag{4}
$$

*where $\|\cdot\|_F$ denotes the Frobenius norm.*

*Proof.* Recall that the second fundamental form has the following properties:

$$
\mathbf{II} = \begin{bmatrix} \mathbf{x}_{uu}^T \mathbf{n} & \mathbf{x}_{uv}^T \mathbf{n} \\ \mathbf{x}_{vu}^T \mathbf{n} & \mathbf{x}_{vv}^T \mathbf{n} \end{bmatrix} = \begin{bmatrix} \mathbf{x}_u^T \mathcal{D}(\mathbf{n})\mathbf{x}_u & \mathbf{x}_u^T \mathcal{D}(\mathbf{n})\mathbf{x}_v \\ \mathbf{x}_v^T \mathcal{D}(\mathbf{n})\mathbf{x}_u & \mathbf{x}_v^T \mathcal{D}(\mathbf{n})\mathbf{x}_v \end{bmatrix} = \mathbf{B}^T \mathcal{D}(\mathbf{n})\mathbf{B},
$$

where $\mathbf{B} = [\mathbf{x}_u \ \mathbf{x}_v]$ and $\mathcal{D}(\cdot)$ is the total derivative of surface normal $\mathbf{n}$ in canonical coordinate system. With this properties, then we can express the fundamental form of surface $\mathcal{M}_f$ as $\mathbf{II}_f = \mathbf{B}^T \mathbf{J}_D^T \mathcal{D}(\mathbf{n}_g)\mathbf{J}_D\mathbf{B}$. Then the difference between two fundamental forms will be:

$$
\mathbf{II}_g - \mathbf{II}_f = \mathbf{B}^T \left(\mathcal{D}(\mathbf{n}_g) - \mathbf{J}_D^T \mathcal{D}(\mathbf{n}_f)\mathbf{J}_D\right) \mathbf{B}.
$$

For any vector $[a_1 \ b_1]$ and $[a_2 \ b_2]$, define $\mathbf{t}_{1,2} = \mathbf{B}[a_{1,2} \ b_{1,2}]^T$. Note that $\mathbf{t}_{1,2}$ are tangent vectors for surface $\mathcal{M}_g$ since $\mathbf{t}_{1,2}^T \mathbf{n}_g = a_{1,2}\mathbf{x}_u^T \mathbf{n}_g + b_{1,2}\mathbf{x}_v^T \mathbf{n}_g = 0$. As a result, $\mathbf{t}_{1,2} = \mathbf{P}_g \mathbf{t}_{1,2}$.

$$
\begin{aligned}
[a_1 \ b_1] \left(\mathbf{II}_g - \mathbf{II}_f\right) [a_2 \ b_2]^T &= [a_1 \ b_1]^T \mathbf{B}^T \left(\mathcal{D}(\mathbf{n}_g) - \mathbf{J}_D^T \mathcal{D}(\mathbf{n}_f)\mathbf{J}_D\right) \mathbf{B}[a_2 \ b_2]^T \\
&= \mathbf{t}_1^T \left(\mathcal{D}(\mathbf{n}_g) - \mathbf{J}_D^T \mathcal{D}(\mathbf{n}_f)\mathbf{J}_D\right) \mathbf{t}_2 \\
&= \mathbf{t}_1^T \mathbf{P}_g^T \left(\mathcal{D}(\mathbf{n}_g) - \mathbf{J}_D^T \mathcal{D}(\mathbf{n}_f)\mathbf{J}_D\right) \mathbf{P}_g\mathbf{t}_2 = 0.
\end{aligned}
$$

Since this holds for any $a_{1,2}$ and $b_{1,2}$, the matrix $\mathbf{II}_g - \mathbf{II}_f$ must be zero. $\qquad\square$

## 3.2 Connection to Thin-shell energy

In this section, we will show the connection between thin-shell energy and our model. Recall that thin-shell energy encourages the materials to resist stretching and bending, thus creating elastic deformation. The thin-shell energy is a combination of two terms: the stretching energy, which measures how the surface (and in particular the tangent planes) has been stretched, and the bending energy, which measures the degree to which the curvatures along different directions have changed.

When the surfaces has a natural parameterization $\mathbf{x}(u, v)$ and $\mathbf{y}(u, v)$, this shell energy is written as following:

$$
\mathcal{E}(\mathbf{x}, \mathbf{y}) = \iint \underbrace{\lambda_s \left\|\mathbf{I}_\mathbf{x}(u, v) - \mathbf{I}_\mathbf{y}(u, v)\right\|_F^2}_{\mathcal{E}_s(\mathbf{x},\mathbf{y})} + \underbrace{\lambda_b \left\|\mathbf{II}_\mathbf{x}(u, v) - \mathbf{II}_\mathbf{y}(u, v)\right\|_F^2}_{\mathcal{E}_b(\mathbf{x},\mathbf{y})} \, dudv, \tag{5}
$$

where $\mathbf{I}_f$ and $\mathbf{II}_f$ denotes the first and second fundamental forms ($\mathbb{R}^{2\times 2}$ matrices) for surface $f(u, v)$, and $\|\cdot\|_F$ is the Frobenius norm. Intuitively, the first fundamental form describes lengths and areas, so the difference $\|\mathbf{I}_\mathbf{x} - \mathbf{I}_\mathbf{y}\|_F^2$ measures stretching. The second fundamental form describes curvatures,

so the difference $\|\mathbf{II_x} - \mathbf{II_y}\|_F^2$ measures bending. $\lambda_s$ and $\lambda_b$ control the resistance to stretching and bending, respectively.

In the previous section, we've established that if the stretch loss reaches 0, then the difference between the first fundamental forms will reach 0. This means that $\mathcal{E}_s$ in the thin shell energy will reach 0 as well. Similarly, if the bending loss reaches 0, then the difference between the second fundamental forms will be 0. As a result, $\mathcal{E}_b$ will reach 0.

# 4 Derivation for Change of Variable

In this section, we will provide derivation for Equation 8 in Section 5.3.3 of the main paper. We will first compute the surface area change and then leverage the change of surface area to derive Equation 8.

**Theorem 4.1.** *(Change of surface area) Assume the iso-surface $\mathcal{M}_f$ of field $f$ is parameterized by $\mathbf{y}(u, v) \in \mathbb{R}^3$. Field $g$ is defined by $g(\mathbf{x}) = f(D(\mathbf{x}))$ with an invertible function $D : \mathbb{R}^3 \to \mathbb{R}^3$. Let $\mathbf{x} = D^{-1}(\mathbf{y})$, and let $n_g$ be the surface normal of $\mathcal{M}_g$ at certain point. Then we will have the change of local surface area to be*

$$\frac{|\mathbf{y}_u \times \mathbf{y}_v|}{|\mathbf{x}_u \times \mathbf{x}_v|} = \det\left(\mathbf{P}_g \mathbf{J}_D^T \mathbf{J}_D \mathbf{P}_g + n_g n_g^T\right) \tag{6}$$

*Proof.* By the identity of cross product, we have

$$|\mathbf{x}_u \times \mathbf{x}_v| = \det \begin{bmatrix} \mathbf{x}_u^T \mathbf{x}_u & \mathbf{x}_u^T \mathbf{x}_v \\ \mathbf{x}_v^T \mathbf{x}_u & \mathbf{x}_v^T \mathbf{x}_v \end{bmatrix} = \det \mathbf{I}_g = \det\left(\mathbf{B}^T \mathbf{B}\right),$$

where $\mathbf{B} = [\mathbf{x}_u \ \mathbf{x}_v]$. Similarly we have:

$$\begin{aligned} |\mathbf{y}_u \times \mathbf{y}_v| &= \det \begin{bmatrix} \mathbf{y}_u^T \mathbf{y}_u & \mathbf{y}_u^T \mathbf{y}_v \\ \mathbf{y}_v^T \mathbf{y}_u & \mathbf{y}_v^T \mathbf{y}_v \end{bmatrix} \\ &= \det \begin{bmatrix} \mathbf{x}_u^T \mathbf{J}_D^T \mathbf{J}_D \mathbf{x}_u & \mathbf{x}_u^T \mathbf{J}_D^T \mathbf{J}_D \mathbf{x}_v \\ \mathbf{x}_v^T \mathbf{J}_D^T \mathbf{J}_D \mathbf{x}_u & \mathbf{x}_v^T \mathbf{J}_D^T \mathbf{J}_D \mathbf{x}_v \end{bmatrix} \\ &= \det \mathbf{I}_f = \det\left(\mathbf{B}^T \mathbf{J}_D^T \mathbf{J}_D \mathbf{B}\right), \end{aligned}$$

Now defined an extended version of $\mathbf{B} \in \mathbb{R}^{3 \times 3}$ as $\mathbf{B}_n = [\mathbf{x}_u \ \mathbf{x}_v \ n_g]$. Then we will have:

$$\det\left(\mathbf{B}_n^T \mathbf{B}_n\right) = \det \begin{bmatrix} \mathbf{x}_u^T \mathbf{x}_u & \mathbf{x}_u^T \mathbf{x}_v & \mathbf{x}_u^T n_g \\ \mathbf{x}_v^T \mathbf{x}_u & \mathbf{x}_v^T \mathbf{x}_v & \mathbf{x}_v^T n_g \\ n_g^T \mathbf{x}_u & n_g^T \mathbf{x}_v & n_g^T n_g \end{bmatrix} = \det \begin{bmatrix} \mathbf{B}^T \mathbf{B} & \mathbf{0} \\ \mathbf{0}^T & 1 \end{bmatrix} = \det\left(\mathbf{B}^T \mathbf{B}\right)$$

With this, we will show the following:

$$\begin{aligned} \det\left(\mathbf{B}^T \mathbf{J}_D^T \mathbf{J}_D \mathbf{B}\right) &= \det\left(\mathbf{B}_n^T \mathbf{B}_n\right) \det\left(\mathbf{P}_g \mathbf{J}_D^T \mathbf{J}_D \mathbf{P}_g + n_g n_g^T\right) \\ &= \det\left(\mathbf{B}_n^T \left(\mathbf{P}_g \mathbf{J}_D^T \mathbf{J}_D \mathbf{P}_g + n_g n_g^T\right) \mathbf{B}_n\right) \\ &= \det\left(\mathbf{B}_n^T \mathbf{P}_g^T \mathbf{J}_D^T \mathbf{J}_D \mathbf{P}_g \mathbf{B}_n + \mathbf{B}_n^T n_g n_g^T \mathbf{B}_n\right). \end{aligned}$$

First, observe that

$$\begin{aligned} \mathbf{P}_g \mathbf{B}_n &= \left(\mathbf{I} - n_g n_g^T\right) \mathbf{B}_n \\ &= \mathbf{B}_n - n_g n_g^T [\mathbf{x}_u \ \mathbf{x}_v \ n_g] \\ &= \mathbf{B}_n - n_g [n_g^T \mathbf{x}_u \ n_g^T \mathbf{x}_v \ n_g^T n_g] \\ &= [\mathbf{x}_u \ \mathbf{x}_v \ n_g] - n_g [\mathbf{0} \ \mathbf{0} \ 1] \\ &= [\mathbf{x}_u \ \mathbf{x}_v \ n_g] - [\mathbf{0} \ \mathbf{0} \ n_g] \\ &= [\mathbf{x}_u \ \mathbf{x}_v \ \mathbf{0}] \end{aligned}$$

With this we have:
$$\mathbf{B}_n^T\mathbf{P}_g^T\mathbf{J}_D^T\mathbf{J}_D\mathbf{P}_g\mathbf{B}_n = [\mathbf{x}_u\ \mathbf{x}_v\ \mathbf{0}]^T\mathbf{J}_D^T\mathbf{J}_D[\mathbf{x}_u\ \mathbf{x}_v\ \mathbf{0}]$$
$$= \begin{bmatrix} \mathbf{x}_u^T\mathbf{J}_D^T\mathbf{J}_D\mathbf{x}_u & \mathbf{x}_u^T\mathbf{J}_D^T\mathbf{J}_D\mathbf{x}_v & 0 \\ \mathbf{x}_v^T\mathbf{J}_D^T\mathbf{J}_D\mathbf{x}_u & \mathbf{x}_v^T\mathbf{J}_D^T\mathbf{J}_D\mathbf{x}_v & 0 \\ 0 & 0 & 0 \end{bmatrix}$$
$$= \begin{bmatrix} \mathbf{B}^T\mathbf{J}_D^T\mathbf{J}_D\mathbf{B} & \mathbf{0} \\ \mathbf{0}^T & 0 \end{bmatrix}$$

Similarly, since we know $n_g^T\mathbf{B}_n = [n_g^T\mathbf{x}_u\ n_g^T\mathbf{x}_v\ n_g^T n_g] = [0\ 0\ 1]$, we have:
$$\mathbf{B}_n^T n_g n_g^T\mathbf{B}_n = (\mathbf{B}_n n_g)^T (\mathbf{B}_n n_g)$$
$$= [0\ 0\ 1]^T[0\ 0\ 1]$$
$$= \begin{bmatrix} 0\ 0\ 0 \\ 0\ 0\ 0 \\ 0\ 0\ 1 \end{bmatrix}.$$

Putting these two terms together, we have:
$$\det\left(\mathbf{B}_n^T\mathbf{P}_g^T\mathbf{J}_D^T\mathbf{J}_D\mathbf{P}_g\mathbf{B}_n + \mathbf{B}_n^T n_g n_g^T\mathbf{B}_n\right) = \det\left(\begin{bmatrix} \mathbf{B}^T\mathbf{J}_D^T\mathbf{J}_D\mathbf{B} & \mathbf{0} \\ \mathbf{0}^T & 1 \end{bmatrix}\right) = \det\left(\mathbf{B}^T\mathbf{J}_D^T\mathbf{J}_D\mathbf{B}\right).$$
$\square$

**Theorem 4.2.** *(Change of variable intergration)* Assume the iso-surface $\mathcal{M}_f$ of field $f$ is parameterized by $\mathbf{y}(u,v) \in \mathbb{R}^3$. Field $g$ is defined by $g = f(D(x))$ with an invertible function $D : \mathbb{R}^3 \to \mathbb{R}^3$. $\mathcal{M}_g$ can be parameterized by $\mathbf{x}(u,v) = D^{-1}(\mathbf{y}(u,v))$. Then the surface intergral can be evaluated by:
$$\int_{\mathbf{x}\in\mathcal{M}_g}\mathcal{L}(\mathbf{x})d\mathbf{x} = \int_{\mathbf{y}\in\mathcal{M}_f}\mathcal{L}(D^{-1}(\mathbf{y}))\left|\det\left(\mathbf{J}_D\mathbf{P}_g + n_f n_g^T\right)\right|^{-2}d\mathbf{y}. \tag{7}$$

*Proof.* We first apply the change of variable for surface intergral:
$$\int_{\mathbf{x}\in\mathcal{M}_g}\mathcal{L}(\mathbf{x})d\mathbf{x} = \iint\mathcal{L}(\mathbf{x}(u,v))|\mathbf{x}_u\times\mathbf{x}_v|dudv$$
$$= \iint\mathcal{L}(D^{-1}(\mathbf{y}(u,v)))\frac{|\mathbf{x}_u\times\mathbf{x}_v|}{|\mathbf{y}_u\times\mathbf{y}_v|}|\mathbf{y}_u\times\mathbf{y}_v|dudv$$
$$= \int_{\mathbf{y}\in\mathcal{M}_f}\mathcal{L}(D^{-1}(\mathbf{y}))\left(\frac{|\mathbf{y}_u\times\mathbf{y}_v|}{|\mathbf{x}_u\times\mathbf{x}_v|}\right)^{-1}d\mathbf{y}$$
$$= \int_{\mathbf{y}\in\mathcal{M}_f}\mathcal{L}(D^{-1}(\mathbf{y}))\left(\det\left(\mathbf{P}_g\mathbf{J}_D^T\mathbf{J}_D\mathbf{P}_g + n_g n_g^T\right)\right)^{-1}d\mathbf{y}.$$
Then we apply the fact that $\det\mathbf{A}^T = \det\mathbf{A}$, we have:
$$|\det\mathbf{J}_D\mathbf{P}_g + n_f n_g^T|^{-2} = |\det\left(\left(\mathbf{J}_D\mathbf{P}_g + n_f n_g^T\right)^T\left(\mathbf{J}_D\mathbf{P}_g + n_f n_g^T\right)\right)|^{-1}$$
$$= |\det\left(\mathbf{P}_g^T\mathbf{J}_D^T\mathbf{J}_D\mathbf{P}_g + \mathbf{P}_g^T\mathbf{J}_D^T n_f n_g^T + n_g n_f^T\mathbf{J}_D\mathbf{P}_g + n_g n_f^T n_f n_g^T\right)|^{-1}$$
$$= |\det\left(\mathbf{P}_g^T\mathbf{J}_D^T\mathbf{J}_D\mathbf{P}_g + n_g n_g^T + \mathbf{P}_g^T\mathbf{J}_D^T n_f n_g^T + n_g n_f^T\mathbf{J}_D\mathbf{P}_g\right)|^{-1}$$

Now it's left to show that $\mathbf{P}_g^T\mathbf{J}_D^T n_f n_g^T = n_g n_f^T\mathbf{J}_D\mathbf{P}_g = 0$. Recall that $\mathbf{J}_D^T n_f = Cn_g$ where $C$ is a constant $C = \|\mathbf{J}_D^T n_f\|$. Also $\mathbf{P}_g^T = \mathbf{P}_g$. With these, we have:
$$\mathbf{P}_g^T\mathbf{J}_D^T n_f n_g^T = C\mathbf{P}_g n_g n_g^T$$
$$= C\left(\mathbf{I} - n_g n_g^T\right)n_g n_g^T$$
$$= C\left(n_g n_g^T - n_g n_g^T n_g n_g^T\right)$$
$$= C\left(n_g n_g^T - n_g n_g^T\right) = \mathbf{0}$$
Since $\left(n_g n_f^T\mathbf{J}_D\mathbf{P}_g\right)^T = \mathbf{P}_g\mathbf{J}_D^T n_f n_g^T = \mathbf{0}$, so $n_g n_f^T\mathbf{J}_D\mathbf{P}_g = \mathbf{0}$. $\square$

# 5 Implementation details

## 5.1 Architectures

**Input network.** The input network is MLP with sin activation [6]. For Armadillo, Cactus, Dino, and Half-Noisy Sphere, the hidden dimensions are 3-512-512-512-512-512-512-3. For simpler shapes (i.e., Cylinder and Bar), the hidden dimensions are 3-512-512-512-3. For the 2D rectangle, we use 2-128-128-128-128-2. For shape smoothing and sharpening, the output network assumes the same topology as the input network.

**Deformation network.** For all our deformation experiments, we use six invertible residual blocks. The positional encoding of residual blocks uses $L = 5$, so the input coordinate with $d$-dimensions will be expanded to $d(L + 1)$-dimension before being processed by the MLP. These MLPs use ELU [1] activations with $256$ hidden nerons. Specifically, the dimensions evolves as following: $d$-$d(L + 1)$-256-256-$d$. We apply spectrum normalization to each of the linear layers to enforce Lipschitz continuity.

## 5.2 Optimizations and hyper-parameters

**Input network** We use the following objectives to train the input neural field $F_\theta$:

$$\mathcal{L}(\theta) = \int_{\mathbf{x} \in U} (F_\theta(\mathbf{x}) - \mathbf{y}_{gtr})^2 + k_g (\|\nabla_{\mathbf{x}} F_\theta(\mathbf{x})\| - 1)^2 d\mathbf{x}. \tag{8}$$

We set $k_g = 0.01$ according to Gropp et al. [4]. We use Adam with a $1e - 5$ learning rate to train for $300000$ steps to obtain the input network. For simpler shapes, we also do early stopping when we see the training loss converged before using up all the steps.

**Distillation** We choose $\lambda_g = 0.01$ and $\lambda_c = 0.0001$ for all experiments. For smoothing, we set $\beta = 0$; for sharpening, we set $\beta = 2$. The threshold is chosen to be 50 for all shapes. Each updating iterations, we will sample 5000 uniformly from the space the input is well supervised (e.g., in our case, it is $[-1, 1]^3$). We use Adam optimizer with a learning rate of $1e - 5$ to train for 1000 iterations and report the performance for the last iterations. These experiments were run on a workstation GPU (e.g., Geforce 1080 Ti or TitanX, with about 10-12 GB memory) for about 10 minutes.

**Deformation** For deformation, we initialize $\lambda_0 = 100$. We use Adam optimizer for the SGD step using a learning rate of $1e - 5$. Similarly, we set $\mu = 1e - 5$ to update $\lambda_i$. We set $\tau_c = 1e - 4$ for all handles. For deformation experiments, we use slightly different hyper-parameters (i,e. $k_s$ and $k_b$) for different shapes. For 3D shapes, we set $k_s = 0.1$ and $k_b = 1e - 3$. This hyper-parameter usually works well for most shapes. For cylinders, we tune down the bending resistance to $k_b = 1e - 5$ to better preserve surface area. Each experiment was run on a workstation GPU (e.g., Geforce 1080 Ti or TitanX, with 12GB memory) for about 5 - 10 hours. We ran for 100k iterations but usually will early stop when training loss converges.

## 5.3 Experiments

**Data processing pipeline.** We use package *mesh-to-sdf*[2] to produce ground truth SDF for 5M uniformly sampoled points (within $[-1, 1]^3$) and for 5M near surface points. The near-surface points are obtained by first uniformly sampled 5M points on the surface, then perturbed these points by $\epsilon \sim \mathcal{N}(\mathbf{0}, \sigma^2 \mathbf{I})$ with $\sigma = 0.1$. These ground truths will be used as training. This data preprocessing pipeline requires about 15-30 minutes to run. Together with the training, it will take about just as long as optimizing for the deformation objectives (i.e., about 10 hours).

**Shape filtering baselines.** To extract meshes for the baselines, we first evaluate the field on $512^3$ voxels and run marching cube to obtain high-poly meshes. Then we run Taubin filter [8] on these high-poly meshes directly to produce the filtered mesh for baseline. For the low-poly baseline, we will run quadratic decimation [3]. We tune the target number of polygons according to the particular shapes such that it can best preserve detail while working well with Taubin smooth to be performed

---

[2] https://github.com/marian42/mesh_to_sdf

later. It's true that our processing pipeline for the extracted meshes is far from perfect. Nevertheless, the mesh pipeline requires additional processing steps to work with the extracted meshes. This indicates that they might not be the best candidate to manipulate geometries represented with neural fields.

**Shape deformation baselines.** Similar to the shape filtering baselines, we first run marching cube to extract high-resolution meshes. Then we will apply ARAP [7] algorithm implemented in Open3D [9] to process the mesh. We set the smoothing hyper-parameter to be $1000$ and optimize for enough steps until the energy functions converged (the difference between iterations is less than $1e-5$).

For the additional baseline with remeshing, we run quadratic decimation to simply the marching cubed extracted mesh into one that has roughly the same number of faces as the input mesh (i.e., the original one used in ARAP paper). As for SR-ARAP baseline, we set the regularization loss weight to be $1e-3$.

# 6 Additional quantitative results

## 6.1 Quantitative Results for Deformation

This section provides quantitative measures of how our method and the baseline preserve surface area and volume. In table 1, we report the change of surface area and the change of the volume in percentage before and after transformation. We can see that the baseline optimize solely for preserving the surface area while sacrificing the volume preservation. On the other hand, our method tries to balance the change of surface area and change of volume. As a result, the deformation of our method seemed more natural when the shape of interests was assumed to have volume. The baseline

Table 1: Change of surface area and volum for different deformation operations.

|  | Area (%) | | Volume (%) | |
| --- | --- | --- | --- | --- |
| Operation | Base | Ours | Base | Ours |
| Bar twist | 0.96 | 1.01 | 14.72 | 2.04 |
| Cylinder rotate | 0.08 | 13.91 | 21.05 | 15.59 |
| Cylinder translate | 0.19 | 6.68 | 37.30 | 8.39 |
| Cactus rotate | 0.07 | 3.77 | 10.97 | 4.67 |
| Cactus bend | 0.04 | 3.51 | 10.86 | 3.96 |
| Cactus translate | 0.13 | 1.28 | 28.47 | 0.87 |
| Armidillo side | 0.11 | 0.35 | 4.93 | 0.78 |
| Armidillo bend | 0.05 | 0.84 | 3.40 | 2.00 |
| Armidillo back | 0.04 | 1.31 | 12.15 | 1.07 |
| Dino bend | 0.04 | 1.29 | 1.89 | 1.57 |

method's deformation is more suited for the shape that behaves like a thin shell without volumes. Such behavior can potentially be achieved by our method by tuning hyper-parameters $\lambda_s$ and $\lambda_b$.

## 6.2 Quantitative Results for Sampling

Table 2: CD-ratio, EMD-ratio, and time reported in Table 1 of the original paper. We include the standard deviation in parenthesis.

|  | Dino | | Armadillo | |
| --- | --- | --- | --- | --- |
| Metrics | Base | Ours | Base | Ours |
| CDr | 1.54(0.05) | **1.04 (0.01)** | 1.36 (0.03) | **1.02 (0.02)** |
| EMDr | 3.38 (0.55) | **1.15 (0.15)** | 3.30 (0.44) | **1.08 (0.1)** |
| Time | **0.15 (0.03)** | 0.21 (0.06) | **0.12 (0.05)** | 0.18 (0.03) |

In addition, we provide the standard deviation of 10 different runs for sampling results in Table 2. The number suggests that the difference is much larger than the standard deviation, showing statistical significance of the improvement in terms of CD ratio and EMD ratio.

### 6.3 Visualization of Threshold-ed Points for Shape Smoothing and Sharpening

In Section 4, we suggest filtering out points with high curvature values before computing the curvature regularization for sharpening. Note that this curvature-based filtering is a heuristic we use to deal with the issue that the isosurface of neural fields represented using SIREN architecture has lots of high noise. As we can see in Figure 1 of the main paper, even a planar isosurface (with curvature expected to be 0) contains points with much higher curvature. With that said, the curvature estimation using the second derivative of SIREN is not always accurate. While it might seem mild in Figure 1, we can run into points with extremely high curvature. Such points will cause training divergence. This curvature-based filtering schema aims to filter out points with high curvature to stabilize training. Depending on the application, we usually tune the threshold such that about 80% of the points remain after filtering. Figure 1 shows the points that got filtered out.

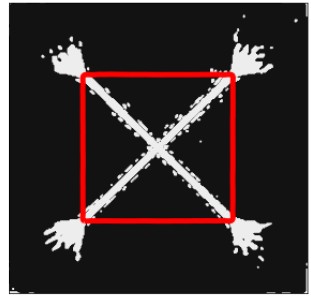

Figure 1: White points are filtered out while computing the curvature regularization.

## 7 Discussion of Limitations and Future Work

This section will provide a more detailed discussion of the limitation and show how it can potentially be addressed in future work.

**Optimizing speed too slow.** It takes about 5 to 10 hours to optimize a deformation using a single GPU (e.g., Geforce 1080Ti) with 12GB memory. This prevents our algorithms from being used interactively. To improve optimization speed, one can potentially decrease the size of the deformation network (i.e., reducing to a single residual block) and use progressive training.

**Output is not an SDF.** Right now, the output field of the deformation step is not an SDF, while the input requires the field to be an SDF. This prevents the algorithms from being used multiple times. To address this issue, future work can either develop *reinitialization* techniques. For example, we can finetune the network to fulfill the Eikonal constraints without breaking the iso-surface. Such techniques have been studied in prior work such as Osher and Fedkiw [5]. Alternatively, we can reduce the requirement of input from a valid SDF in a bounded domain to merely a neural field whose iso-surfaces are differentiable. Another way to achieve it is to add Eikonal constraints loss during training.

**Shape filtering method does not reuse the existing network.** The current shape filtering method finetunes on the input network. This method either perturbs the original network. Alternatively, we can reinitialize a network with the same architecture, initialize it with the input network's parameter, and optimize it with the filtering objective. This will create a new computational graph that does not contain the original network. Such an approach will lose the edit history, and it is memory intensive. These issues can potentially be fixed by using a correction network to obtain the filtering operation [2].