# OpenReview forum: "Geometry Processing with Neural Fields"
_NeurIPS.cc/2021/Conference — NeurIPS 2021 Poster_

### Official Review · Reviewer_7RWH · 2021-07-09

**Rating:** 9
**Confidence:** 3

**Summary:**

The paper explores the use of neural implicit fields for classical geometry processing operations. In doing so, it addresses the two fundamental issues that make this non-trivial (1) the identification of losses that do not require local parametrization domains over which integration of differential properties is usually performed, and (2) the sampling of neural implicit fields on the surface for the evaluation of such losses.

**Ethical Concerns:**

None.

**Limitations And Societal Impact:**

Limitations stated, but should be moved to the main paper.

I had a good chuckle when I read the societal impact paragraph (thank you authors!). I think it's important for NeurIPS to have this sections for some papers (e.g. generative face/text models, face replacement, etc...) but asking it from every papers (e.g. this one) is just plain silly. This is, of course, a commentary towards NeurIPS, and not towards the authors.

**Main Review:**


I am extremely excited about this paper, and had hoped to work on this myself if I had enough human resources available. However, it turns out it was actually more difficult than I expected (see comments in my summary), and the authors did an excellent job at addressing the technical challenges it presented. So overall, I am extremely happy about the idea pitch, overall convinced by the method section (this is because I must admit I do not have the time to triple check the derivations in the supplementary! which is also why I mark my review with lower confidence), and mostly satisfied with the evaluations (see below). Overall, I would expect this to be a capstone paper that will attract thousands of citations in the future, despite all the limitations the authors correctly point out, and I am excited to see it accepted.

I only have a few remarks/questions for the authors, which I hope can be addressed, but that I do not think would affect my rating (the key factor in championing/rejecting the paper is the correctness of the derivations, to which I ought to delegate to another reviewer... or the AC).


## core questions
1. In my first read, I was taken by surprise by technical "jumps", only to realize the derivations were in the supplementary... it would be an excellent idea to explicitly refer to sub-sections in the supplementary in the main paper (and I hope the two will be concatenated in a likely future arXiv submission!)
1. You compare ARAP-like deformations on meshes for marching cubes. However, MC meshes have elements with terrible finite-element conditioning, and nobody in his right mind would execute polygonal mesh processing algorithms on such input. I think it would be a good idea to include comparisons to isotropically re-meshes version of MC meshes (Botsch et al.) alongside the "raw" MC comparison you currently have?
1. L91: I think you are missing a relevant work in NIFs deformation, especially as it's related to the invertibility you employ in your current solution → "ShapeFlow: Learnable Deformations Among 3D Shapes" @ NeurIPS 2020
1. L150-L153: I am not quite convinced about this argument, which is made even worse by the fact that there is no ablation/experiment to verify its effectiveness. Are you doing this because, since you are executing some neural form of "explicit Euler" mean-curvature flow, this results in instability in the smoothing operation during optimization? This is particularly important as "sharp" meshes processed "Implicit Fairing" work with no problems, which reveals a flaw/weakness in your algorithm?
1. Sec5.3: what's "mixing time"? also, with a perfect SDF (9) would converge in a single iteration, so why would that be a problem? Further, I am a bit confused about this section. My understanding is that SDF-projections of ambient samples would overall results in an over-sampling of large positive curvature areas, and an under-sampling of large negative curvature areas (which Fig.2 seems to confirm). However, I do not recall this being discussed in the text, and the "rejection" sampling you provide is a simple (yet somehow effective) "hack" to resolve this problem. Further, I think you are missing comparisons showing that Langevin on the *deformed* NIF does not work (both with and without rejection sampling?).
1. L275: I think you must justify why it is necessary to employ this "constraint satisfaction" threshold, I am not fully convinced why it would be necessary. Also, why did you not use the penalty method instead of AG, did the simpler penalty method just fail, or was it too hard to tune?
1. Fig.5: I am not quite sure what L_s+L_b is... isn't that the full model? And if so, why does it perform so poorly? Because you picked wrong loss-balance hyperparameters? (I was expecting something better than L_b)

## minor points
1. L120: I was confused by this statement, as $P$ does not measure tangential stretch, it's a tool that combined with other quantities (later in the paper) allows you to understand tangential stretch
1. L135(equation number missing): I would strongly advise not to use $k$ and $\kappa$, as my eyes crossed while reading your math. Could you use $\lambda$ or $\omega$ as a symbol to balance different losses?
1. Would be a lot better if the results for the smoothing section were presented right away... especially as it's just one figure and ~10 lines of text?
1. I would much rather read the limitations in the main paper than in the supplementary material -- it is one of the most important sections of any paper! It would be more tasteful to move the background recap on differential geometry to supplementary instead.
1. What does the "r" in CDr and EMDr mean?

**Time Spent Reviewing:**

4

---

> ### Author Response · Authors · 2021-08-11
> **Thank you for the positive review! We will respond to your core questions in this comment**
>
> Thanks for the positive feedback! We are glad that you find the idea exciting!
>
> ---------------
>
> ### Q2: baselines with remeshing
>
> We agree that the mesh directly extracted from MC might not be suitable for ARAP deformation. In fact, this may contribute to the volume distortion artifacts in the ARAP baseline. This [link](https://www.dropbox.com/s/evpcke9jpq7vej7/NeurIPS%202021%20Rebuttal%20AdditionalResults.pdf?dl=0) contains additional qualitative comparison to a baseline that apply some basic remeshing techniques on the MC extracted mesh before applying to ARAP. It seems like remeshing will be able to alleviate some of the volume distortion artifacts.
>
> While it’s possible that one can construct a pipeline with remeshing to improve ARAP’s performance on MC outputs from NIFs, this further shows that the geometry processing algorithm is tied to the discretization quality of the surface (as pointed out by **reviewer 1jPU**). Our algorithm optimizes NIFs directly, thus able to circumvent the need to discretize the surface. We will update the paper to include these additional results and discussions.
>
> ---------------
>
> ### Q4: smoothing algorithm doesn’t optimize on high curvature locations
>
> The reason why we need to filter out high-curvature points is that the estimation of curvature using the second derivative of the SIREN network is not accurate. As we can see from Figure 1, the ostensibly smooth surface which we expect to have 0 curvature turns out to be very noisy when zoomed in. Such high-frequency noise is also different from the sharp mesh since it exists everywhere in the field. Without proper filtering of the points, the training will diverge.
>
> Our curvature-based filtering is a heuristic to deal with the noise caused by SIREN architecture. It will be an interesting research direction to search for a more principled approach to resolve such an issue. For example, it will be an interesting future direction to improve the SIREN architecture to eliminate such issues. It’s also very interesting to connect such an objective with explicit Euler mean curvature flow to develop a more stable version of the filtering technique.
>
> ---------------
>
> ### Q5: questions regarding sampling
> Our goal is to generate point clouds uniformly from the zero iso-surface of the NIF. Similar to prior work  [A], we formulate the problem as sampling from a distribution using a Langevin dynamic. Specifically, we iteratively perturb the points with Gaussian noise and then do SDF projection. Under this framework, “mixing time” refers to the number of iterations taken for the distribution of sampled points to approach the desired distribution (uniform over the surface). Single SDF projection will lead to point clustering in a high curvature area, which creates a distribution that’s far from the desired one. Prior works [A, B] have shown that running Langevin dynamic with more steps can alleviate such a problem, but the number of iterations will be prohibitively high without any coarse-to-fine strategy (i.e. “mixing time” is long). Our proposed technique is essentially a coarse-to-fine strategy that allows efficient sampling of the surface.
>
> The following table compares sampling quality on the deformed mesh of the dino example. Sampling in the deformed field without rejection sampling doesn’t work. Sampling with rejection in the deformed field can provide a high-quality sample for cases when SDF is not distorted severely after deformation. Since the deformed field is constantly changing during training, sampling from the deformed NIFs will require us to resample constantly. Sampling from the original mesh can avoid such an issue.
>
> |        | Original mesh |           | Deformed mesh |           |
> |--------|---------------|-----------|---------------|-----------|
> | Metric | No Rej.       | With Rej. | No Rej.       | With Rej. |
> | CDr    |          1.54 |      1.04 |   1.38 |      1.06 |
> | EMDr   |          3.37 |      1.15 |   4.53 |      1.15 |
>
> The table reports CDr/EMDr, which refers to CD/EMD-ratio. It's computed by $CDr(A, B) = \frac{CD(sampe(A), sample(B))}{CD(sample(A), sample(A))}$ and similarly for EMDr. The division will normalize the CD/EMD to account for the surface area of the shape. Both metrics are lower the better.
>
> ---------------
>
> ### Q6: justification of Augmented Lagrangian method and thresholding
>
> We found that we can also achieve good results without the Augmented Lagrangian method or the thresholding. One way to do that is to leverage the penalty term as mentioned in the question. Note that the optimization method is not related to the core contribution of the paper. We will update the text to clarify this point.
>
> ---------------
>
> ### Q7: comparison between L_b, L_s, and L_b + L_s
> We agree that the reason why L_b + L_s model doesn’t look quite as good is that it has the wrong hyperparameters. The goal of the task is to produce a “deformation that resembles the natural behavior of real objects” (L159). L_b-only model produces deformation similar to materials that are extremely resistant to bending, while L_s-only models materials resistant to stretching. Users can tune the parameter to find the right balance between stretching and bending that resembles the material of interest. What’s shown in the ablation is a model trained with randomly picked hyper-parameters (k_s=1, k_b=1e-3) to demonstrate the model is capable of producing a mixture. As a result, such deformation might not resemble any common real-world material.
>
> ---------------
>
> ### Q3: missing important reference ShapeFlow
>
> ShapeFlow models the deformation field using CNFs (an invertible network) and provides several theoretical insights of such invertible deformation fields, but ShapeFlow focuses on a different task, learning deformation fields between two known meshes. We will update the text to address all mentioned missing references.
>
> ---------------
>
> ### Q1 and other writing issues
>
> Thanks for all the comments regarding writing! We will update the text to address all of the issues mentioned.
>
> ---------------
>
> [A] Cai, Ruojin, et al. "Learning gradient fields for shape generation." Computer Vision–ECCV 2020: 16th European Conference, Glasgow, UK, August 23–28, 2020, Proceedings, Part III 16. Springer International Publishing, 2020.
> [B] Song, Yang, and Stefano Ermon. "Generative modeling by estimating gradients of the data distribution." NeurIPS (2019).

---

### Official Review · Reviewer_1jPU · 2021-07-14

**Rating:** 6
**Confidence:** 4

**Summary:**

This submission introduces methods to smooth, sharpen and deform shapes represented as neural implicit fields (NIF) without meshing them first.
Smoothing and sharpening are done by training a new NIF with the same SDF values and the proposed regularization loss to increase or decrease curvature.
Deformations are obtained by applying an invertible deformation field that warps 3D space to match deformation constraints. In this regard, the contributions of the paper are:
- Representing the deformation field as an invertible network, with positional encoding
- Introducing 2 regularization terms to preserve shape properties: stretching and bending losses, that respectively penalize change of dot product and curvature on the surface.
- Introducing a uniform sampling method on the shape surface to apply these stretching and bending losses.

**Limitations And Societal Impact:**

The submission is honest about a main limitation of the proposed method: it is slow and does not allow for interactive editing. The supplementary material reports a 5 to 10 hours optimization time for deforming a mesh. This should maybe be reported in the main paper, along with the time required by the baseline ARAP.
Overall, it would be valuable to present the entire Discussion section of the supplementary material in the main paper.

Societal impact is correctly assessed.

**Main Review:**

__Originality:__ the presented toolbox is novel and allows to manipulate 3D shapes from their implicit representations. Related works on neural implicit representations of 3D shape are reported, and none of them allows to manipulate shapes without having to mesh them first. In this perspective, the proposed method is novel.


__Significance:__ the *overall aim of the paper not entirely clear* to me: what is the advantage of processing shapes in their implicit representations? The abstract sells the idea that “Meshes are hard to optimize for topology change“, and optimizing a NIF should indeed allow for topology change. But this is neither demonstrated in the paper, nor re-emphasized later. So the **abstract is a bit misleading.**

The introduction then advertises the use of “higher order derivatives” for deforming shapes, but again this is not clear how or when.
Overall, as stated at the end of Section 2, the presented “method avoids iso-surface extraction”, but what is the benefit of doing so? It also seems that iso-surface extraction is anyways required to get the shape in an explicit form, be it before or after processing.
The real value of the presented method to me is the fact it decorrelates meshing resolution from shape manipulation parameters, as illustrated from results in 6.1. More emphasis should be placed on this.

Similarly, on line 163, the difference between the proposed method compared to [22, 39, 53] could be stressed for a more impactful submission.

To increase the significance of this work, a demonstration of how it can be applied to different NIF architectures (not only SIRENS) and in the case of a learned shape prior (ex. DeepSDF, ie. apply regularization at training time) would be welcome.

Another concern is the evaluation against baseline method ARAP is not consistent with the originally reported figures in [65]. In Fig. 7 of [65] the cactus is not undergoing the undesirable volumetric deformation presented here and bends smoothly. How can this be explained?

__Quality:__ The ablation study successfully demonstrates the effectiveness of all the proposed components, and visual results are convincing. It is difficult to provide metrics to evaluate deformation and smoothing quality. However,  Tab. 1 from the supplementary showing volume vs surface preservation is interesting and could probably moved to the main paper.

__Clarity:__ figures and mathematical formalism are neat and well written for easily understanding the proposed method, with the exception of 5.2.1 and 5.2.2. They both introduce a formalism based on an available x(u,v) parametrization to compute matrix B. But since x(u,v) is not available, this formalism is then discarded for another one, which is compatible with NIF representations. But why introduce B and the associated formalism in the first place? It does not seem to help building and intuition, and could even be confusing.

Another point that is not very clear is on line 290 about baselines: why rerun the data preparation pipeline to try the processed mesh back into an NIF? What is the goal of getting an NIF back as output?

Overall, this submission is very complete technically speaking and just enough details are given such that a careful reader could probably re-implement the method.

__Cosmetic remarks:__
- line 115 n_d(x) is used to refer to the “surface normal” of a point that is not on the surface… Shouldn’t n_d(.) rather be defined as the SDF field gradient? So that it has a sense to manipulate it for points not on the surface.
- Line 150 : “curvature regularization” would sound better than “curvature mismatch”
- Line 211: x_u, x_v have not been introduced.
- Line 263: “is achieveD by […] keepING”
In the supplementary:
- Parentheses are messed up in the intro
- Line 87 “inteRgration”
- Line 131 “turths”



**Time Spent Reviewing:**

4.5 hours

---

> ### Author Response · Authors · 2021-08-11
> **Thanks for the constructive review! We will address your concerns in this comment:**
>
> Thank you for the thoughtful review!
>
> ---------------
>
> ### Overall aim of the paper
>
> The overall goal of the paper is to give a proof of concept that geometry processing can be done entirely using NIFs, which have many desirable properties such as compactness, high fidelity, and can be optimized for topological changes. Since prior works have tried to leverage NIFs for geometry processing tasks that involve topological changes (L24 and reference [8, 47, 50, 58]), we instead focus on the tasks that might be challenging for NIFs:  smoothing and topology-preserving deformation (L55-56). It’s true that the abstract doesn’t reflect this, thus becoming misleading. We will update the text to address this issue.
>
> ---------------
>
> ### Higher-order derivative
>
> Higher-order derivatives are used implicitly during optimization. The optimization objective involves 1st (normal) and 2nd (curvature) derivatives. To optimize such an objective with SGD, we thus need to have access to higher-order derivatives (in our case, 3rd). We will revise the corresponding sections to make this point clear.
>
> ---------------
>
> ### Avoid Iso-surface extraction
>
> We agree that the true value of the method is that it doesn’t tie to certain spatial resolution or discretization of the surface. Iso-surface extraction that turns NIFs into meshes requires building surface connectivity (i.e. building triangles), which implicitly commit to certain spatial resolution. Furthermore, visualizing NIFs doesn’t require iso-surface extraction. One can run the Ray Marching algorithm to render NIFs without producing meshes. We will make the corresponding text more clear to reflect these points. Thanks for this constructive feedback!
>
> ---------------
>
> ### Why get NIFs back as output? Why not compare the mesh output from ARAP directly?
>
> One of the reasons why we develop geometry processing algorithms for NIFs is to enjoy NIFs’ nice properties such as compactness and not committing to any spatial discretization. With that said, the task of interest inputs and outputs NIFs so that the entire procedure can potentially avoid discretization of the surfaces. The ARAP’s output mesh is almost identical to the one extracted from NIFs after rerunning the data preparation pipeline. Please refer to this [link](https://www.dropbox.com/s/1wm7ptikrgrwiil/NeurIPS%202021%20Rebuttal%20ARAPMesh.pdf?dl=0) for qualitative results and the response to **Reviewer 29Uu** for more discussion and quantitative results.
>
> ---------------
>
> ### ARAP baseline not consistent with the original figures
>
> As mentioned by **Reviewer 7RWH**, the finite element condition of meshes from the Marching Cube is not necessarily suitable for geometry processing. We test our code on the original meshes in ARAP (which usually are low resolution, about 1-2k faces) and our code is able to reproduce similar performance as figures in ARAP. The same code produces volume distortion artifacts with mesh output extracted using the marching cube algorithm. Such distortion can be less severe when we apply simple remeshing techniques to the extracted meshes. This suggests that mesh geometry processing algorithms are sensitive to discretization quality, while our algorithm doesn’t have such an issue since we convert NIFs to NIFs directly without committing to any particular discretization. Please refer to this [link](https://www.dropbox.com/s/evpcke9jpq7vej7/NeurIPS%202021%20Rebuttal%20AdditionalResults.pdf?dl=0) for qualitative result and our response to **Reviewer 7RWH** for more discussion. We will update the experiment section to include these qualitative comparisons.
>
> ---------------
>
> ### Apply to different architecture
>
> We expect our method can be applied to architectures similar to SIREN (i.e. with non-trivial higher-order derivatives), such as the Fourier Feature network [A]. We will include experiments with other NIFs architectures in the next iteration.
>
> ---------------
>
> ### General comments of writing
>
> Thank you for providing such constructive feedback. We will revise the related work section to emphasize the core impact as suggested. We will also reorganize section 5.2 to be more readable and other writing comments.
>
> ---------------
>
> [A] Tancik, Matthew et al. “Fourier Features Let Networks Learn High Frequency Functions in Low Dimensional Domains.” NeurIPS (2020).

---

> > ### Comment · Reviewer_1jPU · 2021-08-23
> > **Thank you! About ARAP -> Mesh -> NIF -> Mesh**
> >
> > Thank you for the detailed answer.
> > This is a minor concern, but I am still not entirely convinced by the relevance of the (ARAP -> Mesh -> NIF -> Mesh) pipeline. Considering the task at hand is shape smoothing (or deformation), simply doing (ARAP -> Mesh) seems to be a valid baseline to me. The (Mesh -> NIF -> Mesh) part of the current pipeline looks superfluous.

---

> > > ### Author Response · Authors · 2021-08-24
> > > **Thanks! We will update the baseline.**
> > >
> > > We really appreciate the feedback! We understand your concern and we will include ARAP output as a baseline in the future version.

---

### Official Review · Reviewer_8TN2 · 2021-07-15

**Rating:** 6
**Confidence:** 3

**Summary:**

This work introduces a framework for geometry processing on surfaces represented as neural networks (neural implicit fields).
Namely, authors design two algorithms: one for surface smoothing, and one for surface deformation with control points (deformation handles). The core idea behind both is to introduce a new neural network (for smoothing another neural implicit field and for deformation an "invertible deformation field"), and optimize it with gradient descent, using a set of losses defined directly on implicit surface. Qualitative experimental evaluation is conducted on a set of meshes, for deformation comparison is provided to a single baseline: as-rigid-as-possible deformation, and for smoothing to Taubin smoothing.

**Limitations And Societal Impact:**

Authors shortly discuss limitations in their work, including the slowness due to the fact that the method relies on optimization, which indeed can cause difficulties for interactive editing as well as applicability of the method as a backbone for downstream tasks.  However, since the practical implications of those can be very severe, a more extensive formal evaluation and more context in the form of comparisons and timings (on similar hardware) would be necessary for readers to fully understand the scope of these issues.

**Main Review:**

### Writing quality / clarity

**+** The paper is really well written: there is enough background provided to make the manuscript self-contained, and proposed methods are discussed in sufficient detail and explained well.

**+/-** Authors also do a good job at referring to existing work on neural implicit representations, but omit some of important references for mesh deformations (e.g. [1]-[4]).

### Contributions / significance

**+** On the one hand, this line of work is certainly worth investigating: neural implicit representations have been used successfully for geometry modeling as well as for some other relevant applications (e.g. neural rendering), so having reliable methods for surface smoothing and deformation could be quite useful.

**-** On the other hand, it is doubtful that the methods presented in this work are at all practical: they rely on heavy optimization, and effectively require training an additional neural network for each deformation / smoothing. This, as authors themselves admit in their limitations paragraph, probably makes interactive use of proposed method highly impractical. Considering that most of the applications would either be real-time / interactive or use mesh smoothing / deformation as a basic building block (e.g. when sampling novel shapes for shape optimiziation), it is unlikely that the framework presented in this work can actually be used in practice. Moreover, in some applications (again, e.g. shape optimization), it is often very useful for the underlying parameterization or deformation algorithm to be differenitable, which seems to be hard to achieve when the deformation itself is a heavy optimization process.

**-** Neural implicit representations can be used to represent a _collection_ of shapes ([22], [25]), in which case conditioning signals or latent space can be used as an optimizable parameter space and similar constraints can be used as an objective. I do not find the argument (L97) "it’s unclear how to apply these methods to out-of-distribution shapes" very convincing, because proposed method in a way is even less generalizable, as it is only operating on a single sample.

**+** Formulation of the stretching and bending losses for implicit surfaces from Section 5.2 will probably be interesting to the community.


### Evaluation

There are multiple concerns that I have about the evaluation.

**-** There are several alternatives to ARAP (see references below for non-exhaustive list, there is also RBF), some of which more recent, which are not reported by the authors. Moreover, results of ARAP seem to be very different from the original manuscript (e.g. on the same "cactus" shape). More generally, it is a bit hard to judge the quality of the algorithm from such a limited number of static examples and few baseline comparisons.

**-** It would be interesting to see how methods that directly operate on implicit surfaces work on these tasks (some of those referred to on L88-L92).

**-** The weakest point of the proposed framework is how expensive it is. Yet, there is no comparison in terms of speed or memory requirements with respect to the baselines.

**-** (minor) I am not very convinced that it is necessary to compare to the surfaces automatically extracted from the NIF, rather than original meshes: especially since for most of the reported examples they are available.

**-** (minor) It was a bit strange that authors claimed the benefits of implicit representations as being able to optimize for topology changes, and then report results on topology-preserving deformations.


### References
[1] Botsch et al '2006: "Primo: coupled prisms for intuitive surface modeling”

[2] Chao et al '2010: “A simple geometric model for elastic deformations”

[3] Crane et al '2011: "Spin Transformations of Discrete Surfaces"

[4] Levi et al '2014: "Smooth rotation enhanced as-rigid-as-possible mesh animation"

[5] de Boer et al '2007: "Mesh deformation based on radial basis function interpolation"



**Time Spent Reviewing:**

3

---

> ### Comment · Reviewer_7RWH · 2021-08-11
> **Don't let the perfect be the enemy of the good**
>
> Apologies, but I need to champion this submission
>
> 1) is the paper technically wrong, or technically trivial?
> I see no mentions of this in your review
>
> 2) does it try to do something that was never attempted before?
> as far as I can tell, any GP on neural implicits would require meshing, including MeshSDF
>
> 3) is it important to compare to every single fairing/deformation paper in history?
> not necessarily... how does this help making the point that classical GP ops can be executed directly on neural implicits? Further, ARAP is one of the most widely accepted standards (likely the most relevant)
>
> 4) Is it slow? of course! But can it be made faster by follow-up works?
> most likely
>
> 6) Can it be generalized to leverage shape collections?
> authors claim not, but it'd be possible
>
> So overall your "limitations / reasons for rejection" look to me like inspiration for future works!

---

> ### Author Response · Authors · 2021-08-11
> **Thanks for the thoughtful review! We will address your concerns in details in this comment**
>
> Thanks for your thoughtful review! We are happy that you find our paper “interesting to the community” and the task “worth investigating”. The main goal of our paper is to provide **a proof of concept** that geometry processing can be done **entirely using NIFs**. While there are couple of decades worth of research into geometry processing for meshes, there is currently no known analog that can be used directly on NIFs. In fact, it is not even clear from the literature if geometry processing using only NIFs is even possible. Our main contribution is to show that this is in fact possible. This necessitates a novel network architecture, loss function, and sampling scheme to deform and smooth NIF surfaces without producing a mesh. While it is true that eventually we want these algorithms to be practical and real-time, we believe that showing it is even possible is a necessary first step. We hope that our work will inspire future, more practical, algorithms.
>
> ---------------
>
> ### Speed
>
> We report in the supplementary the time and memory needed to perform the deformation algorithm. We are happy to move this discussion to the main paper.
>
> As mentioned in supplementary L183-184, we can apply progressive learning to speed up the algorithm. If the edit is large, our approach indeed will take several hours. But when these algorithms are used by artists, they usually specify a trajectory along with the handles deformed. This trajectory has also been leveraged by prior work (ARAP). Using this trajectory we can substantially reduce the computational cost. For example, if we break down the large deformation into 10 small deformations, we are able to learn a single invertible residual layer with 32 hidden units to optimize each of the deformations **within 100 iterations** (**32 seconds** to achieve **<0.5%** area and volume distortion).
>
> Another potential way to speed up is suggested by **Reviewer 29Uu**. We can learn a prior over the deformation fields from a collection of data. During test time, we can sample from such prior to provide a good initialization to improve convergence time. This idea has been applied to speed up training NIF, allowing creating high-quality NIF within just a couple SGD steps [A, B].
>
> In addition, with future development of hardware, optimization techniques, and systems, it’s entirely possible that we can obtain a real-time geometry processing algorithm.
>
> ---------------
>
> ### Comparing with existing methods that deform NIFs + method that represents a collection of shapes (L88-99)
> These prior works are very relevant as they are operating on NIFs. But they are not able to solve the problem of deforming an arbitrary shape based on user input.
>
> These methods roughly fall into two categories. The first category [11,16,34,39,49,53,54,56] focuses on predicting deformation between two shapes. The target shape is used as input for the algorithm. While this is a very interesting task, this isn’t the focus of our paper. The second category [22, 25, 26, 29, 79] learns to represent a collection of shapes (e.g. chairs). Editing can be done by finding a shape from the representable collection that can satisfy the user's input. Generalization to shapes outside the category is challenging. For example, it’s unclear how a model trained on ShapeNet chairs can represent or deform Armadillo. Though our method operates on one shape at a time, the same optimization procedure can be applied to different shapes.
>
> ---------------
>
> ### ARAP results are different from the original paper
>
> The task of interest in our paper is to process a NIF into another NIF. As a result, both the input and output of the algorithm should be NIF. The natural way to use mesh processing algorithms for this task is to apply them to the mesh extracted from NIF using the marching cubes algorithm. As mentioned by **Reviewer 7RWH**, these meshes are not discretized in the best condition for mesh processing algorithms to succeed. This volume distortion seemed to be consistent with what’s reported in SR-ARAP [C]. We show in this [link](https://www.dropbox.com/s/evpcke9jpq7vej7/NeurIPS%202021%20Rebuttal%20AdditionalResults.pdf?dl=0) that our code is able to reproduce ARAP’s result on the mesh they provided, and improving meshing quality does improve performance. This suggests that mesh processing algorithms are usually sensitive to discretization qualities, while our processing algorithm is able to circumvent this difficulty using NIFs that are not committed to certain spatial resolution (also mentioned by **Reviewer 1jPU**). We will include these additional results and discussions in the updated version of the paper.
>
> ---------------
>
> ### More comparison
>
> We choose ARAP as the baseline because it’s the SOTA algorithm that optimizes the thin-shell energy, which is the same type of energy we use for our algorithms. Other algorithms, such as RBF or Poisson, optimize slightly different objectives [9]. One can potentially develop loss functions for NIF that correspond to those alternative objectives using similar techniques as our paper. We include results with SR-ARAP [C], a follow-up ARAP variant, in this [link](https://www.dropbox.com/s/evpcke9jpq7vej7/NeurIPS%202021%20Rebuttal%20AdditionalResults.pdf?dl=0). Note that our method shows comparable performance as the SR-ARAP results.
>
> The goal of our paper is not to surpass SOTA geometry processing, but rather to provide a proof of concept to show that it’s possible to use NIFs to achieve comparable results with SOTA mesh processing algorithms. As mentioned by **Reviewer 1jPU**, the important value of our algorithm is to achieve geometric processing without committing to certain surface discretization. We will modify the text to make the experiment claims and settings more clear and include results with more ARAP variants.
>
> ---------------
>
> ### Comparing with mesh outputted from ARAP
> The mesh output from ARAP is qualitatively very similar to those output from its corresponding NIF. Please see the response to **Reviewer 29Uu** for more details.
>
> ---------------
>
> ### Missing reference and writing issues
> Thanks for pointing out such issues! We will incorporate this feedback in the next iteration.
>
> ---------------
>
> [A] Sitzmann, Vincent, et al. "Metasdf: Meta-learning signed distance functions." NerIPS (2020).
>
> [B] Tancik, Matthew et al. “Learned Initializations for Optimizing Coordinate-Based Neural Representations.” CVPR (2020).
>
> [C] Levi, Zohar, and Craig Gotsman. "Smooth rotation enhanced as-rigid-as-possible mesh animation." IEEE transactions on visualization and computer graphics 21.2 (2014): 264-277.

---

> > ### Comment · Reviewer_8TN2 · 2021-08-23
> > **Thanks for such a detailed answer!**
> >
> > Thanks for such a detailed answer! Below are some minor comments on the rebuttal.
> >
> > *Speed:* I think it is definitely worth discussing this in the main body of the paper (and provide an actual timings); otherwise, a reader might get a wrong idea about how far the method is from being practical.
> >
> > *Comparing ... collection of shapes:* generalization is indeed challenging, but the point is that you can always fine-tune the network and/or add additional optimization and smoothness constraints, which (arguably) seems trivial and leads to a very similar procedure to what you propose in this work.
> >
> > *ARAP results are different from the original paper*: I see your point, and it is worth doing an ablation on this. I do think it is reasonable to expect a comparison to an actual baseline that would be used in practice.
> >
> > In summary, considering that other reviewers found this work interesting, and assuming that authors do include additional discussions as promised, I will be inclined to rise my original rating.

---

### Official Review · Reviewer_29Uu · 2021-07-22

**Rating:** 7
**Confidence:** 4

**Summary:**

This paper proposes a method to perform some common operations from geometry processing on shapes represented as neural implicit fields (i.e., deep SDFs) rather than parameterized meshes. In particulars, by utilizing the fact that high order derivatives of NIFs can be computed automatically, the authors formulate several loss terms, which allow them to apply sharpening, smoothing, or deformation to an existing implicit field via a reoptimization. They show qualitative and qualitative results and comparisons to the mesh-based analogs of their algorithm and present a fairly comprehensive ablation study validating their design choices.

**Limitations And Societal Impact:**

Both are adequately discussed in Section 7 as well as in the Supplementary.

**Main Review:**

With deep implicit 3D shape representations becoming more and more expressive and popular, it is important to develop algorithms for making them useful for artists and designers who desire direct control over geometry. The proposed method is a nice first step towards developing geometry processing tools that operate on NIFs rather than meshes. I think the paper is very well written and clearly organized, shows nice experimental results, and motivates a lot of interesting and useful directions for future work. Below are some suggestions and questions about the method and exposition.

I don't entirely understand why the smoothing/sharpening losses are evaluated at points below a curvature threshold (as described in Section 4). If SIREN isosurfaces exhibit so much high-frequency noise, which points remain after the filtering? It would be good to include some clarification about this and possibly a figure illustrating which points get filtered out.

Some of the equations in Section 5.2 are very densely packed in the text and difficult to read---especially, e.g., the inline 2x2 matrix on line 231. It would be okay to move some of these details to the Supplementary Material, but spacing should be added for readability.

While evaluating baselines on meshes extracted from the implicit fields makes some sense, it would also make sense to show the mesh-based methods applied to the original input meshes. Currently, the process of mesh -> implicit field -> mesh is a bit roundabout and not entirely fair, since the proposed method is only evaluated on implicit fields that originate from meshes.

Is it possible to incorporate these geometric operations on NIFs into a data-driven learning pipeline in order to avoid the expensive optimization for each edit? For instance, can some of the same loss functions be used to learn these operations generically rather than on a specific input?

**Time Spent Reviewing:**

3 hours

---

> ### Author Response · Authors · 2021-08-11
> **Thank you for your constructive review! We will address your concerns in this comment:**
>
> Thanks for the thoughtful review and comments!
>
> ---------------
>
> ### Comparing directly with mesh output from ARAP
>
> Since the main goal of our paper is investigating the possibility of doing geometry processing **entirely** using NIFs (L31), the task of interest should take NIF as input and output another NIF. A natural way to leverage traditional mesh processing algorithms to achieve this is to extract mesh from the input NIF, process the extracted mesh into another mesh, then create a new NIF from such mesh.
>
> The mesh output from ARAP (i.e ARAP -> Mesh) is highly similar to the one extracted from its corresponding implicit field (i.e. ARAP -> Mesh -> NIF -> Mesh). Please refer to this [link](https://www.dropbox.com/s/1wm7ptikrgrwiil/NeurIPS%202021%20Rebuttal%20ARAPMesh.pdf?dl=0)  for the visual comparison among these meshes.
>
> The following table shows the CDr and EMDr between the mesh output from ARAP and the one from its corresponding NIF. We report CDr (EMDr) as computed by:
> $ Dr(A, B) = \frac{D(sample(A), sample(B))}{D(sample(A), sample(A))}, D = CD \text{ or } EMD.$
> Lower values indicate better performance. We report the results averaged over 5 runs.
>
> | Experiment      | CDr (std) | EMDr (std) |
> |-----------------|----|-----|
> | Cactus   |  0.998 (0.01)  | 1.042 (0.194)    |
> | Dino            |  1.005 (0.007)  | 0.966 (0.086)    |
> | Armadillo       | 0.999 (0.005)  | 1.017 (0.127)    |
>
> We can see that these two types of meshes are quantitatively and qualitatively similar. In the next version, we will make sure to include qualitative examples of both types of meshes to avoid unfair comparison. We will also update the text to clarify the motivation.
>
> ---------------
>
> ### Curvature threshold for smoothing loss
>
> Curvature-based filtering is a heuristic we use to deal with the issue that the iso-surface of NIF represented using SIREN architecture has lots of high noise. As we can see in Figure 1, even a planar isosurface (with curvature expected to be 0) contains points with much higher curvature. With that said, the curvature estimation using the second derivative of SIREN is not always accurate. While it might seem mild in Figure 1, occasionally we can run into points with extremely high curvature. Such points will cause training divergence. This curvature-based filtering schema aims to filter out points with extremely high curvature to stabilize training. We observed that about 80% of the points remain after filtering.
>
> As we mentioned in the discussion with **Reviewer 7RWH**, it will be an interesting direction to search for more principled ways to handle such issues. For example, one can improve SIREN architecture to provide better curvature estimation. We will update the text to include such discussion and a figure that visualizes the filtered out area.
>
> ---------------
>
> ### Data driven pipeline to improve optimization time
>
> Yes, this can be a very interesting research direction. One way to achieve this is to learn a prior over the deformation fields by doing different deformation. During test time, sampling from such prior can provide a good initialization to speed up the optimization. Similar ideas have been successful to speed up NIF’s training [A, B].
>
> ---------------
>
> ### Equations in 5.2
>
> Thanks for pointing out this concern! We will update the text in the next version to make this section more clear!
>
> ---------------
>
> [A] Sitzmann, Vincent, et al. "Metasdf: Meta-learning signed distance functions." NerIPS (2020).
> [B] Tancik, Matthew et al. “Learned Initializations for Optimizing Coordinate-Based Neural Representations.” CVPR (2020).

---

### Decision · Program_Chairs · 2021-09-27

**Decision:**

Accept (Poster)

**Comment:**

This paper produced a substantial amount of discussion before and after the rebuttal, but in the end the decision was to accept this work.  Some readers (notably reviewer 7RWH) were excited about the vision presented in this work and possibilities for extension, while some others had concerns about efficiency/practicality.

In the camera-ready, please more explicit in discussing the limitations of the work (e.g. highlighting that it takes 5 hours for a task that typically can be done in real-time), as well as provide a fair comparison to the baseline to avoid misleading the readers (e.g. running ARAP on essentially degraded meshes).